# From Solo to Symphony: Orchestrating Multi-Agent Collaboration with Single-Agent Demos

## Abstract

Training a team of agents from scratch in multi-agent reinforcement learning (MARL) is highly inefficient, much like asking beginners to play a symphony together without first practicing solo. Existing methods, such as offline or transferable MARL, can ease this burden, but they still rely on costly multi-agent data, which often becomes the bottleneck. In contrast, solo experiences are far easier to obtain in many important scenarios, e.g., collaborative coding, household cooperation, and search-and-rescue. To unlock their potential, we propose Solo-to-Collaborative RL (SoCo), a framework that transfers solo knowledge into cooperative learning. SoCo first pretrains a shared solo policy from solo demonstrations, then adapts it for cooperation during multi-agent training through a policy fusion mechanism that combines an MoE-like gating selector and an action editor. Experiments across diverse cooperative tasks show that SoCo significantly boosts the training efficiency and performance of backbone algorithms. These results demonstrate that solo demonstrations provide a scalable and effective complement to multi-agent data, making cooperative learning more practical and broadly applicable.

## 1 Introduction

Multi-agent reinforcement learning (MARL) has emerged as a core paradigm for sequential decision making in environments that require coordination (Shoham & Leyton-Brown, 2008; Lowe et al., 2017; Gronauer & Diepold, 2022). By interacting with the environment and receiving feedback, MARL enables agents to learn cooperative policies, providing a principled framework for solving complex decision-making problems such as autonomous driving (Zhang et al., 2024), large-scale network optimization (Stepanov et al., 2024), and collaborative robotics (Tang et al., 2025).

However, compared to single-agent RL, MARL faces inherent challenges (Busoniu et al., 2008; Hernandez-Leal et al., 2019), including dimensionality explosion, coordination difficulty, and environmental non-stationarity. As a result, training joint policies from scratch is often inefficient, much like asking novices to rehearse a symphony without prior practice: difficult, time-consuming, and unlikely to yield good results. This inefficiency poses a major obstacle to applying MARL effectively in practice.

To address these challenges, a growing line of research has explored offline MARL (Pan et al., 2022; Shao et al., 2023; Li et al., 2023; Liu et al., 2024b) and offline-to-online fine-tuning (Zhong et al., 2025). These methods learn from pre-collected task-specific cooperative trajectories to avoid costly exploration, and refine pretrained policies with limited online rollouts when interaction is allowed. More recent studies have attempted to relax the data assumption by leveraging multi-task cooperative datasets (Zhang et al., 2023a; Chen et al., 2024; Liu et al., 2025) or even non-cooperative multi-agent datasets (Wang et al., 2023). These efforts broaden the scope of usable data and represent important progress, but they remain fundamentally tied to multi-agent trajectories.

Actually, in many cooperative problems, there often exists a corresponding solo version whose demonstrations are much easier to obtain and learn from. For example, in collaborative coding (Dong et al., 2025b) a single coder writes a short piece of code, in household cooperation (Kannan et al., 2024) a single robot performs an individual chore, and in search-and-rescue (Cao et al., 2024)

a single drone searches for one target. Although such demonstrations deviate from the target cooperative setting, they are far from useless. Such as in orchestral performance, it is more effective to let each novice player first master the basics of solo play before attempting a full ensemble. Yet, the potential of solo demonstrations to accelerate MARL remains underexplored. This gap motivates an important but underexplored question:

***Can solo demonstrations be effectively leveraged to accelerate the collaborative MARL?***

An affirmative answer to this hypothesis will validate solo data as a scalable and cost-effective resource. This will be instrumental in fostering efficient learning in settings where cooperative data are limited but solo demonstrations are plentiful (Kannan et al., 2024; Cao et al., 2024; Dong et al., 2025b), consequently making MARL a more viable solution for practical applications.

However, addressing this problem is non-trivial and involves two major challenges. The first is *observation mismatch*: differences in observation dimensionality hinder the direct transfer of solo demonstrations to multi-agent training (Hu et al., 2021; Zhang et al., 2023a; Liu et al., 2025; Yu et al., 2025). In some cases, a single local observation may even correspond to multiple distinct solo views, creating ambiguity for policy reuse. The second is *domain shift*: unlike multi-agent data, whether joint or agent-specific, that inherently encode cooperation (Wang et al., 2023), solo data contain no such information. In addition, discrepancies in environment dynamics between solo and cooperative settings (e.g., individual attributes and observation noise) further exacerbate the gap. These challenges hinder direct policy transfer, highlighting the need to distill knowledge from solo demonstrations and integrate it into cooperative learning. Recently, PegMARL (Yu et al., 2025) has explored leveraging personalized demonstrations to guide MARL training. However, it mainly operates via individual reward shaping, making it difficult to directly apply to most centralized-training–with–decentralized-execution (CTDE) methods, which typically rely on a central critic and a shared team reward. Moreover, it does not naturally extend to settings with multiple solo views.

To tackle these challenges, we propose **Solo-to-Collaborative RL** (SoCo) framework, which transfers knowledge from solo demonstrations to cooperative MARL. SoCo first pretrains a shared solo policy from solo demonstrations via imitation learning, providing a common skill prior for all agents. During cooperative training, local observations are decomposed into solo views aligned with the demonstrations, allowing the reuse of the solo policy to obtain candidate actions. Then, a policy fusion module selects and refines these actions for each agent, adapting them to the cooperative setting and mitigating domain shift. Specifically, inspired by MoE (Cai et al., 2025) and action fusion in single-agent RL (Dong et al., 2025a), a learnable gating selector chooses the most suitable candidate, while an action editor refines it for effective cooperation. This design not only tackles the challenge of solo-to-cooperative transfer but also provides flexibility for task-specific customization.

We validate SoCo across diverse cooperative benchmarks, and the results show that it markedly improves training efficiency while achieving competitive or superior performance. These findings highlight the potential of solo demonstrations as a scalable resource for cooperative MARL.

Our main contributions are summarized as follows:

- We investigate an important yet underexplored problem of leveraging solo demonstrations to benefit cooperative MARL, and show that such data, though lacking explicit cooperative information, can substantially accelerate multi-agent training.

- We develop Solo-to-Collaborative RL (SoCo), a framework for solo-to-cooperative transfer. It decomposes local observations to reuse a pretrained shared solo policy, and employs a policy fusion module trained from cooperative interactions that combines a gating selector for choosing solo actions with an action editor for refining them, enabling more efficient cooperation.

- We validate SoCo on cooperative benchmarks with diverse characteristics and difficulty, showing that it effectively addresses observation ambiguity and domain shift, boosts training efficiency, and achieves competitive or superior performance, highlighting the potential of solo demonstrations as a scalable resource for MARL.

## 2 PRELIMINARY

### 2.1 MULTI-AGENT REINFORCEMENT LEARNING

We model multi-agent reinforcement learning (MARL) within the decentralized partially observable Markov decision process (Dec-POMDP) framework (Oliehoek et al., 2016). Formally, a Dec-POMDP is defined as $\mathcal{M} = \langle \mathcal{N}, \mathcal{S}, \mathcal{A}, \mathcal{O}, P, R, \gamma \rangle$, where $\mathcal{N} = \{1, \ldots, N\}$ denotes the set of agents, $\mathcal{S}$ is the global state space, and $\mathcal{A}$ and $\mathcal{O}$ are the joint action and observation spaces, each formed from the agents' local action $\{\mathcal{A}_i\}_{i=1}^N$ and observation spaces $\{\mathcal{O}_i\}_{i=1}^N$. At each time step, every agent receives a local observation generated from the current global state. Based on the joint action $\mathbf{a} = (a_1, \ldots, a_N)$, the environment transitions to the next state according to $P$, and a shared reward $R(s, \mathbf{a})$ is returned. The goal of MARL is to learn a joint policy $\boldsymbol{\pi} = (\pi_1, \ldots, \pi_N)$ that maximizes the expected discounted return:$J(\boldsymbol{\pi}) = \mathbb{E}_{\boldsymbol{\pi}}[\sum_{t=0}^\infty \gamma^t R(s_t, \boldsymbol{a}_t)]$. The solo case naturally arises when $|\mathcal{N}| = 1$. In this paper, we mainly focus on scenarios with continuous action spaces.

#### 2.1.1 CTDE PARADIGM AND DETERMINISTIC POLICY GRADIENT METHOD

Centralized Training with Decentralized Execution (CTDE) (Oliehoek et al., 2008; Amato, 2024; Li et al., 2025b) is a widely adopted paradigm in cooperative MARL. In CTDE, each agent executes its policy in a decentralized manner, relying only on its own local observation during interaction with the environment. During training, however, the learning process can leverage additional global information (e.g., global states or joint actions) through centralized critics. This design improves training stability and coordination, while keeping execution scalable and realistic.

A representative CTDE algorithm is MADDPG (Lowe et al., 2017). It extends the deterministic policy gradient (DPG) framework to multi-agent settings by introducing a centralized critic for each agent, while keeping actors decentralized. Formally, let agent $i$ have policy $\pi_i(o_i; \theta_i)$ parameterized by $\theta_i$, and the deterministic policy gradient for agent $i$ is:

$$\nabla_{\theta_i} J(\pi_i) \approx \mathbb{E}_{s, \mathbf{a} \sim \mathcal{D}} \left[ \nabla_{\theta_i} \pi_i(o_i) \nabla_{a_i} Q_i(s, \boldsymbol{a}; \psi^i) \big|_{a_i = \pi_i(o_i)} \right],$$

where $Q_i(s, \boldsymbol{a}; \psi^i)$ denotes the centralized critic for agent $i$, parameterized by $\psi^i$. In practice, however, the critics often share a single parameter set $\psi$ across agents, whereas each agent maintains its own policy network.

Building on MADDPG, MATD3 (Ackermann et al., 2019) incorporates the improvements of TD3 (Fujimoto et al., 2018), including twin critics, target smoothing, and delayed policy updates. HATD3 (Zhong et al., 2024) further extends MATD3 by introducing a heterogeneous sequential optimization. In this paper, we focus primarily on the DPG family under the CTDE paradigm, as represented by MATD3 and HATD3. Nevertheless, the proposed framework is, in principle, extendable to stochastic policy methods, such as HASAC (Liu et al., 2024a). We present an extension to HASAC in Appendix E. Moreover, this work primarily focuses on off-policy algorithms. For less sample-efficient on-policy methods (e.g., MAPPO (Yu et al., 2022)), we discuss them in Appendix F.4.

## 3 SOLO-TO-COLLABORATIVE RL

To bridge the gap between solo demonstrations and multi-agent cooperation, and to tackle observation mismatch and domain shift, we propose the Solo-to-Collaborative RL (SoCo) framework. SoCo first learns a shared solo policy from solo demonstrations. Then, during cooperative training, local observations are decomposed into solo views, allowing the reuse of the solo policy to obtain candidate actions. Finally, a per-agent policy-fusion module selects the most suitable candidate policy and refines it for each agent, adapting it to the cooperative setting and mitigating domain shift. We next present each component in detail. The full algorithm is shown in Algorithm 1 in Appendix C.

### 3.1 SOLO POLICY EXTRACTION

A solo policy is learned by imitation from solo demonstrations $\mathcal{D}_s$ and shared by all the agents. For simplicity, our implementation uses standard behavior cloning, minimizing the mean-squared error between the policy's action and the action recorded in $\mathcal{D}_s$ to obtain a deterministic behavioral policy:

$$\min_w \mathbb{E}_{(o,a) \sim \mathcal{D}_s} \big\| \beta_w(o) - a \big\|_2^2. \tag{1}$$

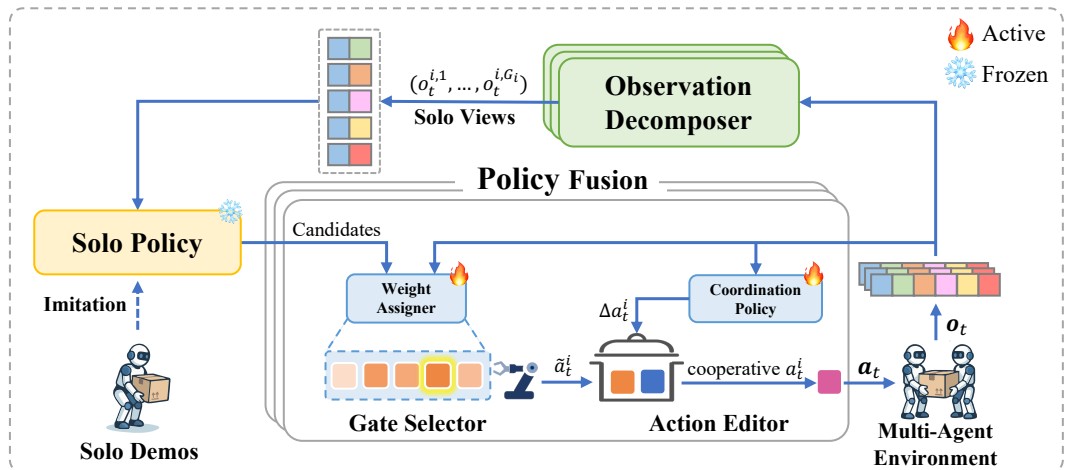

Figure 1: SoCo framework. A shared solo policy is pretrained from demonstrations and kept frozen, then reused through observation decomposition during cooperative training. Coordination ability is injected by the Policy Fusion module, where the Gating Selector selects suitable solo actions and the Action Editor fine-tunes them to mitigate domain shift.

This design choice is flexible rather than mandatory: one could instead adopt a stochastic imitation model that learns $\beta_w(a \mid o)$, for example, parameterizations based on VAE (Kingma & Welling, 2013), diffusion models (Yang et al., 2023), or flow matching (Lipman et al., 2022), by simply switching to a likelihood-based objective without altering the subsequent components of the framework. Finally, the solo policy is shared across agents, and its parameters are frozen during the cooperative learning phase.

### 3.2 OBSERVATION DECOMPOSITION

Given the settings of the cooperative tasks and their corresponding solo tasks considered in this paper, we make a reasonable assumption that the observations in these cooperative tasks are well-defined, structured, and decomposable. Specifically, each observation consists of own features (e.g., velocity, position) and stacked features of all other entities (e.g., teammates or target states). The observation space of the corresponding solo tasks can then be constructed from these feature units (e.g., controlling one HalfCheetah vs. multiple coupled HalfCheetahs).

Hence, following prior works (Liu et al., 2020; Wu et al., 2024; Liu et al., 2025), we introduce a rule-based observation decomposer. Concretely, we decompose the observation of the $i$-th agent at time step $t$, denoted as $o_t^i$, into the self-related component $o_t^{i,0}$ and the entity-related components $\{o_t^{i,k}\}_{k=1}^{K_i}$, where $K_i$ denotes the total number of entities observable by agent $i$. When deployed in cooperative environments, depending on the specific task, we may reassemble the decomposed feature units into $G_i$ valid solo views $\{\tilde{o}_t^{i,k}\}_{k=1}^{G_i}$ for agent $i$ by concatenation, zero-padding, and so on, thereby addressing the issue of inconsistent observation spaces. To more clearly illustrate the observation decomposition process, we provide a concrete example in Appendix D.2.3 for reference.

### 3.3 POLICY FUSION

In the cooperative training phase, the pretrained solo policy cannot be directly transferred. The main obstacles are twofold: (i) a single local observation may map to multiple solo views, producing several candidate actions (e.g., toward different targets) that must be disambiguated; and (ii) domain shift between solo and multi-agent settings often degrades performance, necessitating fine-tuning for effective adaptation.

Therefore, inspired by Mixture-of-Experts (MoE) (Cai et al., 2025) and action fusion techniques in single-agent RL (Dong et al., 2025a), we propose a novel learnable policy fusion module. Notably, our design operates at the agent level and is trained directly on multi-agent samples with standard

MARL optimization, thereby injecting cooperative adaptability into solo policies. Within this module, each agent employs a *Gating Selector* to resolve ambiguity by selecting suitable solo actions, and an *Action Editor* to fine-tune the chosen action for coordination, together enabling effective solo-to-cooperative transfer.

### 3.3.1 GATING SELECTOR

As discussed in Section 3.2, the local observation $o_t^i$ of the $i$-th agent corresponds to $G_i$ solo views $\{\tilde{o}_t^{i,k}\}_{k=0}^{G_i}$. By applying the solo policy, these yield $G_i$ candidate actions:

$$a_t^{i,k} = \beta(\tilde{o}_t^{i,k}), \quad k = 1, \ldots, G_i. \tag{2}$$

However, due to the solo-to-cooperative gap, not all candidate actions are suitable for the current cooperative context, and some may even conflict with each other. To resolve this, SoCo equips each agent with a weight assigner $g_\varphi^i : \mathcal{O}_i \to \mathbb{R}^{G_i}$ that, conditioned on the current local observation $o_t^i$, evaluates the candidate solo actions and assigns weights to them, thereby selecting the most appropriate one for coordination.

To enable learnable action selection, we adopt the Gumbel–Softmax reparameterization (Jang et al., 2017) with the straight-through estimator. The gating weights $g_\varphi^i(o_t^i)$ define a categorical distribution, from which a one-hot action is drawn: the forward pass takes the most probable action, while the backward pass propagates gradients through the soft sample. The resulting action for agent $i$ is:

$$\tilde{a}_t^i = \left\langle \mathrm{GumbelSoftmax}\left(g_\varphi^i(o_t^i)\right), \, \boldsymbol{a}_t^i \right\rangle, \tag{3}$$

where $\boldsymbol{a}_t^i = (a_t^{i,1}, \ldots, a_t^{i,G_i})$ is the set of candidate actions derived from the solo policy.

Moreover, this module is designed to be both general and flexible, allowing adaptation to different scenarios. For instance, the gating function may be rule-based instead of learned, and in the special case of $G_i = 1$, the selector can be omitted entirely.

*Remark: When $G_i$ varies across different observations, a feasible solution for the gating selector is to output a scalar $\mathrm{w}_t^{i,k} = g_\phi^i(o_t^i, \tilde{o}_t^{i,k})$ for each candidate, rather than a vector over all solo views at once. These weights can then be collected and normalized without truncating gradient propagation.*

### 3.3.2 ACTION EDITOR

To leverage the prior knowledge in solo actions while overcoming transfer difficulties from domain shift, we design an action editor that injects cooperative information through residual corrections. Specifically, we introduce a coordination policy $\pi_\theta : \mathcal{O}_i \to \mathcal{A}_i$ that produces a raw residual adjustment to the solo action. To keep this correction bounded and scale-invariant while avoiding gradient saturation, we squash the policy output with $f_L(x) = L \tanh(x/L)$. Given the current local observation $o_t^i$, the adjustment is:

$$\Delta a_t^i = \begin{cases} L \cdot \tanh\left(\dfrac{\pi_\theta(o_t^i)}{L}\right) & \text{if } L > 0 \\ \boldsymbol{0} & \text{if } L = 0 \end{cases} \tag{4}$$

where $L$ is a hyperparameter that controls the strength of the correction. By tuning $L$, the framework can trade off between leveraging solo priors and adapting to multi-agent dynamics.

Then, the final cooperative action is defined as:

$$a_t^i = \mathrm{Clip}\left(\tilde{a}_t^i + \Delta a_t^i\right) \tag{5}$$

where $\tilde{a}_t^i$ denotes the solo action selected by the gating selector, and $\mathrm{Clip}(\cdot)$ is a clipping operator to prevent action overflow. In our implementation, we adopt a $\tanh$-based operator, but the design is modular and allows substituting other operators depending on the task.

### 3.4 COLLABORATIVE POLICY OPTIMIZATION

For notational simplicity, we denote the fused policy as $\Pi_\phi$, where $\phi = \{\varphi, \theta\}$ collects the learnable parameters of the gating selector and action editor. Since SoCo is fully decoupled from the backbone

algorithm, this policy can be optimized with any MARL method. Here we instantiate it with MATD3 (Ackermann et al., 2019) for concreteness.

In MATD3, each agent $i$ maintains two shared centralized critics $Q_{\psi_1}, Q_{\psi_2}$ and an individual actor $\Pi_{\phi_i}$. Given a batch of transition $\mathcal{B} = \{(s_t, \boldsymbol{o}_t, \boldsymbol{a}_t, r_t, s_{t+1}, \boldsymbol{o}_{t+1})\}$, the critic loss is:

$$\mathcal{L}(\psi_j) = \mathbb{E}_{(s_t, \boldsymbol{o}_t, \boldsymbol{a}_t, r_t, \boldsymbol{o}_{t+1}) \sim \mathcal{B}} \left[ \left( Q_{\psi_j}(s_t, \boldsymbol{a}_t) - y_t \right)^2 \right], \quad j = 1, 2, \tag{6}$$

where the target $y_t$ is defined as

$$y_t = r_t + \gamma \min_{k=1,2} Q_{\bar{\psi}_k}(s_{t+1}, \boldsymbol{a}'_{t+1}), \quad (\boldsymbol{a}'_{t+1})_i = \Pi_{\bar{\phi}_i}(o^i_{t+1}) + \epsilon, \tag{7}$$

with $\{\bar{\psi}_k\}_{k=1}^2$ and $\bar{\phi}_i$ denoting target networks and $\epsilon$ being clipped Gaussian noise for policy smoothing. The actors are optimized by maximizing the Q-value estimated by the first critic:

$$\mathcal{L}(\phi_i) = -\mathbb{E}_{(s_t, \boldsymbol{o}_t) \sim \mathcal{B}} \left[ Q_{\psi_1}(s_t, \Pi_{\phi_i}(o^i_t)) \right]. \tag{8}$$

For algorithms with stochastic policies, it suffices to compute the log-probability of the fused action according to Eq. (5) and substitute it into the policy loss.

In this way, SoCo provides a plug-and-play bridge between solo demonstrations and cooperative learning, turning single-agent demonstrations into a scalable and effective complement to multi-agent data, making cooperative learning more practical and broadly applicable. We provide an extension to HASAC (Liu et al., 2024a) to show how SoCo can be combined with a stochastic policy backbone in Appendix E.1. We also discuss the applicability of SoCo to more general settings (e.g., discrete-action environments and unstructured tasks) and the associated challenges in Appendix F.

## 4 EXPERIMENTS

To evaluate the proposed SoCo framework, we conduct experiments on a variety of cooperative tasks. Our goals are to investigate the following questions: (i) Can SoCo improve the sample efficiency of multi-agent algorithms? (ii) Can SoCo enhance the ultimate performance of multi-agent algorithms? (iii) How do the individual components of SoCo contribute to its effectiveness? (iv) How does the quality of solo demonstrations affect the performance of SoCo?

### 4.1 SETUP

**Environments and Tasks.** Following prior works (Sun et al., 2023; Kontogiannis et al., 2025; Zeng et al., 2025), our experiments cover nine tasks from four representative cooperative scenarios: (i) *Spread* (Lowe et al., 2017; Terry et al., 2021), where agents must cover distinct landmarks. This setting introduces target ambiguity but involves little domain shift. (ii) *LongSwimmer* (Peng et al., 2021; de Lazcano et al., 2024), where a multi-segment worm must swim forward, with each agent controlling two consecutive joints. These tasks do not involve target ambiguity but introduce domain shift due to altered dynamics. (iii) *MultiHalfCheetah* (Peng et al., 2021; de Lazcano et al., 2024), where multiple HalfCheetahs are connected in a chain and must run forward together. These tasks avoid target ambiguity but involve noticeable domain shift and present a non-trivial control challenge. (iv) *MultiWalker* (Gupta et al., 2017; Terry et al., 2021), where multiple bipedal robots jointly carry a package forward. These tasks avoid target ambiguity, but are inherently very difficult, with severe domain shift on top of coordination challenges.

Considering the characteristics and difficulty of these environments, we evaluate tasks with 3, 4, and 5 agents in *Spread* and *LongSwimmer*, 2 and 3 agents in *MultiHalfCheetah*, and 2 agents in *MultiWalker*. Details on these environments and tasks are provided in Appendix D.1.

**Data Collection.** To collect solo demonstration data, we first train policies with TD3 (Fujimoto et al., 2018) on the corresponding solo tasks until convergence to an expert level, and then record 1M transition samples. The corresponding solo tasks are: (i) a single agent navigating to one landmark, (ii) a 3-segment worm with 2 joints swimming forward, (iii) a HalfCheetah with altered attributes running forward, and (iv) a single bipedal robot carrying the long package forward. When transferred to multi-agent settings, these lead to (i) goal ambiguity, (ii) domain shift, (iii) notable domain

shift and cooperative difficulty, and (iv) severe domain shift coupled with substantial cooperative difficulty. More details can be found in Appendix D.2.

**Baselines.** We adopt two representative DPG algorithms, MATD3 (Ackermann et al., 2019) and HATD3 (Zhong et al., 2024), and a stochastic policy algorithm, HASAC (Liu et al., 2024a) as our backbone baselines. Their implementations are taken from the HARL codebase (Zhong et al., 2024), on top of which we also build our SoCo variants. The results on HASAC are presented in Appendix E.2. In addition to the *within-backbone* comparison, we also include recent solo-to-multi method PegMARL (Yu et al., 2025) and representative MARL method MAPPO (Yu et al., 2022) in our baselines, enabling a *cross-method* evaluation of these SoCo variants. For the adaptation of PegMARL to our setting and other implementation details, please refer to Appendix D.3.

**Experiment Setup.** SoCo first undergoes imitation learning to obtain the solo policy (5k steps for *Spread*, 100k steps for the others). During cooperative learning, all algorithms perform 10k random interaction steps for warm-up, followed by policy optimization with the same number of environment steps (2M steps for *LongSwimmer* and *MultiHalfCheetah*, 5M steps for *Spread*, and 10M for *MultiWalker*). Except for the correction strength $L$ in SoCo, all hyperparameters are identical to the default settings. For evaluation, each trained policy is tested over 40 episodes, and the average return is reported. All experiments are repeated with 3 random seeds to account for variance. Detailed hyperparameter settings are provided in Appendix D.4.

## 4.2 EVALUATION RESULTS AND ANALYSIS

We evaluate SoCo on nine tasks across four scenarios with varying characteristics and difficulty. Across both backbone algorithms, SoCo improves training efficiency and achieves competitive or superior performance, demonstrating its effectiveness.

**Spread.** In the *Spread* tasks, agents must learn not only to navigate to landmarks but also to resolve target assignment and avoid collisions, with difficulty growing rapidly as the number of agents increases. As shown in Figures 2a–2c, training from scratch becomes highly inefficient under this setting. With SoCo, however, agents first acquire basic navigation skills from solo demonstrations, and during cooperative training, they only need to master target selection and collision avoidance via policy fusion. This significantly improves both training efficiency and final performance. For example, in the 5-agent task, SoCo converges faster and outperforms both backbone algorithms by more than **20%** in final performance, demonstrating its effectiveness in mitigating the challenge of goal ambiguity.

**LongSwimmer.** In the *LongSwimmer* task, agents collaboratively control a multi-segment worm to swim forward. As shown in Figures 2d-2f, although the control difficulty is moderate and both backbone algorithms and SoCo eventually reach similar performance, our framework effectively speeds up training. For example, in the 3-agent task, HATD3-SoCo attains an average return of about 300 at roughly 1.0M steps, while vanilla HATD3 only reaches a comparable level around 1.6M steps, saving nearly **40%** of training samples. These results highlight that SoCo successfully leverages solo demonstrations as a scalable prior to accelerate cooperative learning.

**MultiHalfcheetah.** The *MultiHalfCheetah* task requires multiple HalfCheetah agents connected by elastic tendons to run forward in coordination. The tendon coupling already introduces instability, while the intrinsic difficulty of HalfCheetah control further compounds the challenge. Unlike *Spread* or *LongSwimmer*, this scenario also alters the agents' mass, creating domain shifts that make solo policies non-transferable. Nevertheless, SoCo leverages action editor to adapt solo priors to the shifted dynamics while retaining their basic control skills. As shown in Figures 2g and 2h, this leads to markedly improved training efficiency for the backbone algorithms. In particular, on the 3-agent task, HATD3-SoCo improves the final performance by approximately **83.91%** over the backbone, highlighting the strength of SoCo in leveraging solo demonstrations to boost both efficiency and effectiveness of cooperative training.

**MultiWalker.** The MultiWalker task is the most challenging among the four scenarios. Agents must not only stabilize multiple walkers but also coordinate to carry a long, unstable package under noisy observations. The reward structure is harsh, and the domain shifts are severe, making direct transfer highly difficult. In this setting, backbone MARL algorithms struggle to learn effective package transport within the training budget. By contrast, SoCo leverages policy fusion to refine solo priors

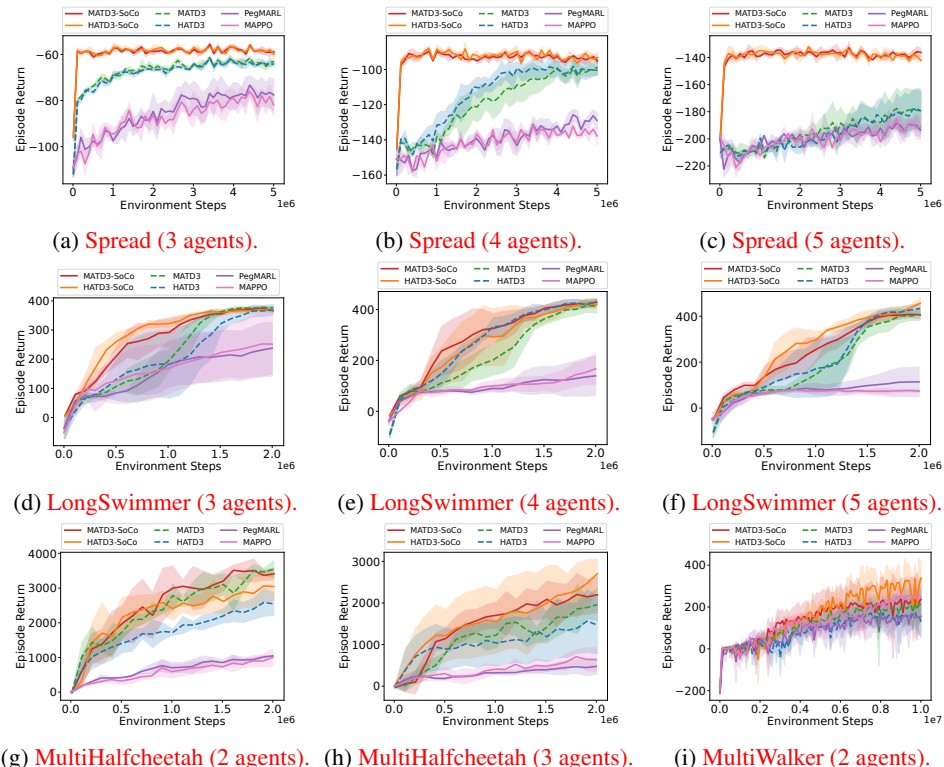

Figure 2: Training curves on nine tasks. Results are averaged over three random seeds, with solid and dashed lines indicating the mean performance and shaded areas representing one standard deviation.

and adapt them to unstable multi-agent dynamics, enabling faster discovery of transport strategies and yielding clear gains in both training speed and final performance. Notably, as shown in 2i, SoCo improves the final performance by **160.71%** on HATD3 and **10.24%** on MATD3 compared to their vanilla counterparts. This shows that SoCo can transfer solo knowledge even under extreme conditions, substantially improving both training efficiency and performance.

**Cross-method Comparison.** The results show that the two DPG-based SoCo variants outperform both PegMARL and MAPPO on almost all tasks (except that, on MultiWalker, MATD3-SoCo is slightly surpassed by MAPPO around 8M steps). This further indicates that the performance gains of SoCo are not only relative improvements over its backbone algorithms, but also competitive compared to other MARL methods. However, it is important to note that this lateral comparison is not entirely fair, since off-policy algorithms are inherently more sample-efficient. In fact, we find that using less sample-efficient on-policy algorithms as the backbone of SoCo can still be challenging. Moreover, as discussed earlier, PegMARL is more suitable for *fully decentralized* methods and not directly applicable to our setting. Nevertheless, it still achieves performance comparable to, or even better than, CTDE-based MAPPO, and we view it as an orthogonal and potentially compatible line of work to SoCo. Please refer to Appendices F.3 and F.4 for a more detailed discussion.

## 4.3 ABLATION STUDY

### 4.3.1 COMPONENT ABLATION

**Gating Selector.** We conduct an ablation study on the 3-agent *Spread* task to isolate the effect of the gating selector. Given the environment structure, setting the correction strength to zero ($L = 0$) already yields strong performance, so we focus exclusively on the gating component. We compare three variants: (i) **Random Gating (RG)**, where targets are sampled randomly at each step; (ii) **Episode-wise Random Gating (ERG)**, where targets are randomly fixed at the start of each episode; and (iii) **Fixed Gating (FG)**, where *distinct* targets are deterministically assigned by agent

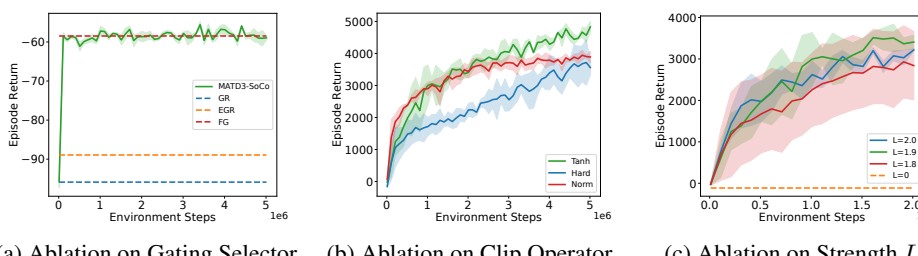

(a) Ablation on Gating Selector.  (b) Ablation on Clip Operator.  (c) Ablation on Strength $L$.

Figure 3: Ablation study of SoCo. Results are averaged over three random seeds, with solid and dashed lines indicating the mean performance and shaded areas representing one standard deviation.

index at the beginning of each episode, serving as an oracle assignment in this scenario. As shown in Figure 3a, randomized gating (RG / ERG) suffers from frequent target conflicts and poor coordination, whereas our learned gating selector can avoid conflicts and perform competitively to FG.

**Clip Operator.** As discussed in Section 3.3.2, we adopt a tanh-based clip operator to prevent fused actions from exceeding valid ranges. Nevertheless, SoCo is designed as a general framework, and different clipping strategies can be customized for specific tasks. To examine this flexibility and assess the suitability of our choice, we conduct experiments on the 2-agent *MultiHalfCheetah* task, evaluating how alternative operators affect both training efficiency and final performance. Using MATD3 as the backbone, we compare two variants:

(i) **Norm**, which normalizes the action as $\text{Clip}(\tilde{a}_t^i + \Delta a_t^i) = (\tilde{a}_t^i + \Delta a_t^i)/(L+1)$;

(ii) **Hard**, which directly truncates actions via $\text{clamp}(\tilde{a}_t^i + \Delta a_t^i, -1, 1)$.

As shown in Figure 3b, the Norm operator accelerates early learning but suffers from weak asymptotic performance, as normalization continuously shrinks the effective action magnitude and reduces policy expressiveness. The Hard operator, on the other hand, truncates actions abruptly, suppressing gradient signals and leading to slow and unstable training. In contrast, our tanh-based design achieves a smoother balance between boundedness and gradient flow, since gradients are only compressed near the action boundaries. This enables both stable learning dynamics and stronger final performance, making the tanh-based operator a natural and effective default choice for SoCo. That said, the framework remains flexible to alternative operators when required by task dynamics.

### 4.3.2 HYPERPARAMETER SENSITIVITY

An important hyperparameter in the SoCo framework is the correction strength $L$, which controls the degree to which the algorithm leverages knowledge from solo demonstrations. We conduct experiments on the 2-agent *MultiHalfCheetah* task with $L \in \{0, 1.8, 1.9, 2.0\}$. Since this environment does not involve multi-goal settings, we can effectively isolate the influence of the gating selector and focus on the impact of this hyperparameter on SoCo's performance. The results in Table 3c show that when $L$ is too small, SoCo relies excessively on solo demonstrations, which limits its training efficiency. In contrast, when $L$ is too large, SoCo adapts quickly to environmental changes in the early stage, but insufficient use of solo knowledge makes it difficult to discover better strategies later, leading to suppressed final performance. Thus, an appropriate choice of $L$ is essential for maximizing the effectiveness of SoCo. In addition, we also examined the case of $L = 0$, where solo policies are directly applied in the multi-agent environment. The results reveal that domain shift prevents the agents from being successfully controlled, underscoring the necessity of policy fusion.

### 4.4 EFFECT OF DEMONSTRATION QUALITY

The quality of solo demonstrations plays a crucial role in the performance of the solo policy, and thus affects both the starting point and the final performance of cooperative training. Therefore, in this section we investigate how demonstration quality influences the performance of SoCo. The experiments conduct on the 2-agent *MultiHalfCheetah* task. Specifically, using the same procedure, we additionally collect solo demonstrations at two quality levels, medium and poor, and train HATD3-SoCo separately on them. In the medium demonstrations, the agent learns stable but slow

locomotion, whereas in the poor demonstrations, the agent fails to run stably. The training results are shown in Figure 4.

Intuitively, poor demonstrations make SoCo rely more heavily on the coordination policy to correct suboptimal behaviors, leading to slower training and lower final performance. Interestingly, with medium demonstrations, SoCo starts with slower initial progress than with expert demonstrations. However, the smoother control pattern better matches the dynamics of the cooperative task (where each agent is effectively lighter), resulting in higher final performance than when pretrained with expert demonstrations.

These results highlight a subtle relationship between solo demonstrations and downstream cooperative training: the expert policy in the solo environment is not always the most beneficial for the cooperative task, and raising an interesting open question: *how should one design solo demonstrations that are best aligned with the downstream multi-agent objective?*

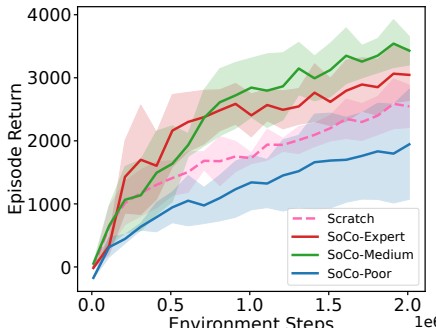

Figure 4: SoCo under different demonstration qualities. Solid and dashed lines indicate the mean performance, and shaded areas represent one standard deviation.

## 5 RELATED WORK

**MARL.** Multi-Agent Reinforcement Learning (MARL) has advanced rapidly, giving rise to diverse paradigms. Fully decentralized methods train and execute policies without centralized information (Tampuu et al., 2017; de Witt et al., 2020), but often suffer from limited coordination. By contrast, the Centralized Training with Decentralized Execution (CTDE) paradigm (Ackermann et al., 2019; Rashid et al., 2020; Yu et al., 2022; Zhong et al., 2024; Li et al., 2025b) has become dominant, enabling centralized training for coordination while preserving decentralized execution. In this paper, we focus on deterministic policy gradient methods under CTDE.

**Transferable MARL.** To mitigate the high cost of training from scratch, transferable MARL aims to reuse experience from source tasks to accelerate learning in target tasks with limited interaction. Existing approaches include offline-to-online (Zhong et al., 2025), multi-task (Chen et al., 2024; Liu et al., 2025; Jha et al., 2025), ad-hoc teamwork (Zhang et al., 2023b; Li et al., 2025a), and mixed-component (Wang et al., 2023). While these methods broaden MARL's applicability, they still assume sufficient data aligned with multi-agent environment. In contrast, exploiting solo demonstrations, abundant yet lacking cooperative signals, remains underexplored. Recent work PegMARL (Yu et al., 2025) attempts to leverage personalized data for individual reward shaping to guide cooperation, but it still suffers from several limitations. Our work further fills this gap, showing that such data can be more effectively leveraged to improve both the training efficiency and performance of cooperative learning.

More detailed discussions are provided in Appendix B.

## 6 CONCLUSION AND FUTURE DIRECTIONS

In this paper, we studied an underexplored problem: how to exploit solo demonstrations to accelerate MARL. We propose a novel Solo-to-Collaborative RL (SoCo) framework, which leverages solo demonstrations by pretraining a shared solo policy and adapting it during cooperative training through policy fusion with a gating selector and an action editor. Experiments across diverse tasks show that SoCo improves training efficiency and achieves competitive or even superior performance, highlighting that solo demonstrations provide a scalable and effective complement to multi-agent data, making cooperative learning more practical and broadly applicable.

This work also opens several avenues for future research, including extending SoCo to heterogeneous demonstrations with skill discovery, applying it to more complex and general environments leveraging large language models, exploring policy fusion in discrete action spaces, investigating the theoretical foundations, and designing more suitable solo demonstrations for cooperative learning.

REPRODUCIBILITY STATEMENT

To ensure reproducibility, we detail our experimental setup in Section 4 and Appendix D, covering environment configuration, solo demonstration collection, implementation details, and hyperparameters. The source code will be made publicly available upon publication.

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

# Technical Appendices

## A  LLM USAGE

In this paper, we use the LLM to polish writing and check grammar issues.

## B  DETAILED RELATED WORK

**MARL.**  Multi-Agent Reinforcement Learning (MARL) has advanced rapidly in recent years, giving rise to diverse paradigms and methods. Fully decentralized approaches train and execute policies without centralized information (Tampuu et al., 2017; de Witt et al., 2020), but their performance is often constrained by the absence of communication among agents. By contrast, the Centralized Training with Decentralized Execution (CTDE) paradigm (Oliehoek et al., 2008; Matignon et al., 2021; Amato, 2024; Li et al., 2025b) has become the mainstream, enabling agents to learn with centralized information for coordination while still executing policies in a decentralized manner. Representative algorithms include HASAC (Liu et al., 2024a), HARL (Zhong et al., 2024), MAPPO (Yu et al., 2022), QMIX (Rashid et al., 2020), and MATD3 (Ackermann et al., 2019). In this work, we adopt deterministic policy gradient methods within the CTDE paradigm, with particular focus on MATD3 and HATD3.

**Transferable MARL.**  Since training MARL from scratch is often sample-inefficient and costly, transferable MARL seeks to reuse experience from source settings to accelerate learning in target tasks with limited additional interaction. Existing approaches span several directions: offline-to-online MARL (Zhong et al., 2025), which leverages offline pretraining to speed up online exploration and correct distributional shift; multi-task MARL (Hu et al., 2021; Wang et al., 2021; Zhang

et al., 2023a; Chen et al., 2024; Liu et al., 2025; Jha et al., 2025), which extracts transferable knowledge from multiple source tasks and applies it to unseen ones; ad-hoc teamwork (Stone et al., 2010; Zhang et al., 2023b; Li et al., 2025a), which exposes agents to diverse teammates to improve robustness when coordinating with unseen partners; and MARL with mixed-component data (Wang et al., 2023), which constructs datasets from individual trajectories generated by different cooperative policies, enriching training diversity while preserving per-step consistency. While these methods broaden the applicability of MARL, they all rely on sufficient multi-agent data. In contrast, the potential of exploiting solo demonstrations, abundant but lacking cooperative signals remains largely unexplored.

Recent work PegMARL (Yu et al., 2025) attempts to leverage personalized data for individual reward shaping to guide cooperation. However, it still suffers from several limitations, such as not being directly applicable to standard CTDE algorithms and being unable to effectively handle settings with multiple solo views. Our work further addresses this gap, showing that such data can be more effectively leveraged to accelerate cooperative training, thereby opening a promising new avenue.

## C  ALGORITHM PSEUDOCODE

---

**Algorithm 1** Solo-to-Collaborative Reinforcement Learning (SoCo)

---

**Input:** Datasets of solo demonstration $\mathcal{D}$ and edit strength $L$.
Initialize the parameters $w$ for solo policy $\beta_w$, $\phi = \{\varphi, \theta\}$ for weight assigner $g_\varphi$ and coordination policy $\pi_\theta$, $\{\psi_j\}_{j=1}^2$ for $\{Q_j\}_{j=1}^2$, and $\bar{\psi}_1, \bar{\psi}_2, \bar{\phi}$ for target networks.
Train solo policy $\beta_w$ with $\mathcal{D}$ according to Eq. (1).
Initialize the replay buffer $\mathcal{B}$.
**for** $i = 1$ **to** $T_{\max}$ **do**
    Obtain the joint observation $\mathbf{o}_t$ from the environment.
    // Agent-wise Solo-to-Collaborative Transfer
    **for** $n = 1$ **to** $N$ **do**
        // Observation Decomposition
        Decompose local observation $o_t^n$ into solo views $\{o_t^{n,k}\}_{k=1}^{G_n}$.
        // Policy Fusion
        Calculate $\boldsymbol{a}_t^n = \beta_w(\{\boldsymbol{o}_t^n\})$ and obtain solo action $\tilde{a}_t^n$ by Eq. (3)
        Calculate editing action $\Delta a_t^n$ by Eq. (4).
        Obtain final action $a_t^n$ by combining $\tilde{a}_t^n$ and $\Delta a_t^n$ according to Eq. (5).
    **end for**
    // Cooperative MARL Training
    Use $\boldsymbol{a}_t = (a_t^1, \ldots, a_t^N)$ to interact with the environment and save $(s_t, \boldsymbol{o}_t, \boldsymbol{a}_t, r_t, \boldsymbol{o}_{t+1})$ into $\mathcal{B}$.
    Sample a batch of transitions $\{(s_t, \boldsymbol{o}_t, \boldsymbol{a}_t, r_t, \boldsymbol{o}_{t+1})\}$ from $\mathcal{B}$.
    Update critics $Q_1$, $Q_2$ and fused policy $\Pi_\phi$ through standard MARL algorithms.
**end for**

---

## D  EXPERIMENTS DETAILS

### D.1  ENVIRONMENTS

We evaluate SoCo on nine tasks across four representative cooperative scenarios:

**Spread (Lowe et al., 2017; Terry et al., 2021).**   As shown in Figures 5a-5c, in this environment, $N$ agents are initialized at random positions in a bounded 2D plane, while $K = N$ landmarks are also randomly placed without overlap. Agents must navigate to distinct landmarks while avoiding collisions. The per-step reward for each agent $i$ is defined as the average of a global and a local component:

$$r_t^i = \tfrac{1}{2}\big(r_t^{\text{global}} + r_t^{\text{local},i}\big).$$

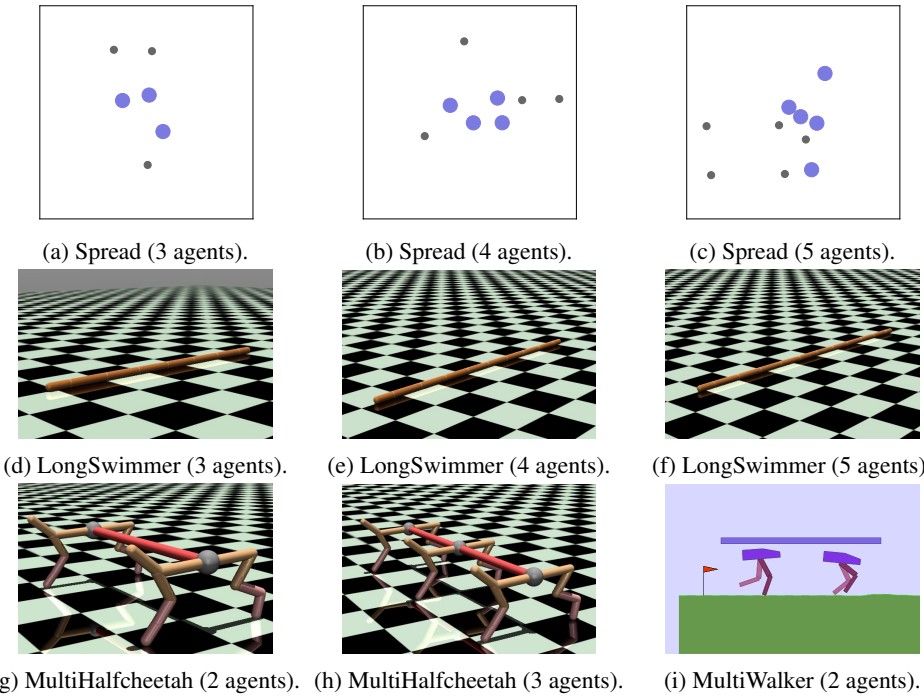

(a) Spread (3 agents).  (b) Spread (4 agents).  (c) Spread (5 agents).

(d) LongSwimmer (3 agents).  (e) LongSwimmer (4 agents).  (f) LongSwimmer (5 agents).

(g) MultiHalfcheetah (2 agents).  (h) MultiHalfcheetah (3 agents).  (i) MultiWalker (2 agents).

Figure 5: All the cooperative tasks in our experiments.

The global reward is shared across agents and encourages coverage of landmarks:

$$r_t^{\text{global}} = -\sum_{k=1}^{K} \min_{j \in \mathcal{N}} \|p_t^j - l_k\|_2,$$

where $p_t^j$ is the position of agent $j$, and $l_k$ is the position of landmark $k$.

The local reward penalizes collisions:

$$r_t^{\text{local},i} = \begin{cases} -C_t^i, & \text{if agent } i \text{ collides with } C_t^i \text{ other agents,} \\ 0, & \text{otherwise.} \end{cases}$$

Finally, the environment reward is the sum over all agents' individual rewards:

$$R_t = \sum_{i \in \mathcal{N}} r_t^i.$$

We evaluate on tasks with 3, 4, and 5 agents.

**LongSwimmer (Peng et al., 2021; de Lazcano et al., 2024).** As shown in Figures 5d and 5f, in this environment, a $(2N + 1)$-segment worm must be controlled to swim forward. Each pair of adjacent segments is connected by a joint, and each agent is responsible for controlling two consecutive joints in sequence. The worm's initial state is sampled from a uniform distribution within a predefined range, while its initial velocity is drawn from Gaussian noise to diversify the dynamics. The per-step reward for each agent $i$ is:

$$r_t^i = v_t - 0.0001 \cdot \sum_{i \in \mathcal{N}} \|a_t^i\|_2^2,$$

where $v_t$ is the forward velocity of the worm, $a_t^i$ is the action taken by agent $i$.

The environment reward is defined as the average of all agents' rewards:

$$R_t = \frac{1}{N} \sum_{i \in \mathcal{N}} r_t^i.$$

We evaluate on tasks with 3, 4, and 5 agents.

**MultiHalfCheetah (Peng et al., 2021; de Lazcano et al., 2024).** As shown in Figures 5g and 5h, in this environment, $N$ HalfCheetah agents are connected in series by elastic tendons and must collaboratively run forward. Each agent's initial state is sampled from a uniform distribution within a predefined range, and its initial velocity is drawn from Gaussian noise to diversify dynamics. The per-step reward for each agent $i$ is:

$$r_t^i = v_t^i - 0.1 \cdot \|a_t^i\|_2^2,$$

where $v_t^i$ is the forward velocity of agent $i$, $a_t^i$ is its action.

The environment reward is defined as the average of all agents' rewards:

$$R_t = \frac{1}{N} \sum_{i \in \mathcal{N}} r_t^i.$$

We evaluate on tasks with 2 and 3 HalfCheetahs.

**MultiWalker (Gupta et al., 2017; Terry et al., 2021).** As shown in Figure 5i, in this environment, $N$ bipedal robots must collaboratively lift and carry a long package forward. The terrain has a randomly undulating profile at the start of each episode. Walkers are initialized at fixed, equally spaced positions in standing poses; to diversify initial conditions, a small random external force is applied to each walker's head at $t = 0$. The package length scales proportionally with the number of walkers, and each walker's observation is corrupted with noise.

At each step, each walker receives a progress reward equal to the forward displacement of the package, plus a small shaping penalty for head tilting and a $-10$ penalty if a walker falls:

$$r_t^i = \Delta x_t^{\text{package}} - 5 \cdot \Delta \theta_t^{\text{head},i} - 10 \cdot \mathbf{1}\{\text{walker } i \text{ falls}\}.$$

Episodes terminate if the package falls, leaves the left edge, or if any walker falls, in which case all walkers receive $-100$. If the package exits the right edge, termination occurs with reward 0.

The environment reward at each step is the sum of individual rewards:

$$R_t = \sum_{i \in \mathcal{N}} r_t^i.$$

We evaluate on task with 2 walkers.

### D.2 SOLO DEMONSTRATION

### D.2.1 DATA COLLECTION

For each cooperative scenario, we first train a policy on its corresponding solo task using TD3 (Fujimoto et al., 2018), and then collect 1M transitions to learn the solo policy. Table 1 summarizes the average episode returns of the solo task demonstrations.

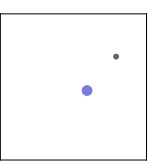 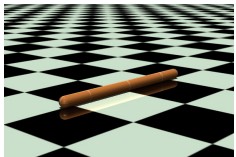 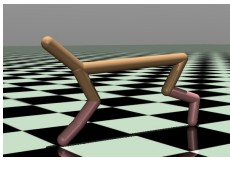 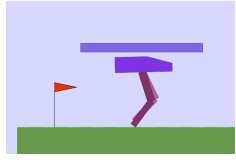

(a) Spread.      (b) LongSwimmer.      (c) MultiHalfcheetah.      (d) MultiWalker.

Figure 6: Solo tasks corresponding to each cooperative scenario.

Table 1: Average episode return of collected solo demonstrations.

| Scenario | Spread | LongSwimmer | MultiHalfcheetah | MultiWalker |
|---|---|---|---|---|
| **Average Episode Return** | -14.77 | 119.44 | 7054.87 | 197.18 |

### D.2.2  SOLO TASKS VS. COOPERATIVE SCENARIOS

These solo tasks, illustrated in Figure 6, exhibit noticeable gaps from their cooperative counterparts, ranging from goal ambiguity (*Spread*), to moderate domain shift (*LongSwimmer*), to notable domain shift and cooperative difficulty (*MultiHalfCheetah*), and to severe domain shift with substantial cooperative difficulty (*MultiWalker*).

Specifically, in the *Spread* scenario, the solo task allows an agent to observe only a single target, whereas in the cooperative setting, multiple targets are visible simultaneously. In the *LongSwimmer*, the motion of the worm is affected by the actions of other agents, introducing a moderate domain shift. In *MultiHalfCheetah*, the solo task doubles the agent's mass and removes tendon constraints, making it simpler than the coupled cooperative case. Finally, in *MultiWalker*, the solo task differs drastically from the cooperative environment: the package length and walker positions change, observations are noisy, and interference from teammates is absent in solo but present in multi-agent training, resulting in severe domain shift and substantially higher cooperative difficulty.

### D.2.3  A SMALL EXAMPLE FOR OBSERVATION DECOMPOSITION

We use the 3-agent *Spread* task as an example to illustrate the observation decomposition process.

In the solo demonstration, there is only one agent and one landmark, so the agent's observation consists of $[own\_pos, own\_vel, landmark\_pos\_r]$, where these components respectively denote the agent's position, velocity, and the relative position of the landmark.

In the 3-agent cooperative environment, each agent additionally observes other agents and landmarks: $[own\_pos, own\_vel, pos\_r\_1, pos\_r\_2, landmark\_pos\_r\_1, landmark\_pos\_r\_2, landmark\_pos\_r\_3, comm\_1, comm\_2]$, where $pos\_r\_i$ is the position of the $i$-th agent relative to itself, $landmark\_pos\_r\_i$ is the relative position of the $i$-th landmark, and $comm\_i$ represents the communication message from agent $i$ (set to 0 in our experiments as communication is disabled). We then decompose each local observation into three solo views:

$$[own\_pos, own\_vel, landmark\_pos\_r\_1]$$
$$[own\_pos, own\_vel, landmark\_pos\_r\_2]$$
$$[own\_pos, own\_vel, landmark\_pos\_r\_3]$$

each of which is passed to the solo policy to generate corresponding action candidates. This illustrative example has been added to the appendix for clarity.

### D.3  IMPLEMENTATION DETAILS

Our implementation and experiments are based on the HARL codebase (Zhong et al., 2024). The additional components introduced by SoCo, i.e., the solo policy, gating selector, and action editor, share the same architecture as the backbone actor network, implemented as 2-layer MLPs with ReLU activations. For action fusion, we adopt a $\texttt{tanh}$-based clip operator: when $\Delta a \equiv 0$, no constraint is applied; otherwise, the fused action is bounded through a $\texttt{tanh}$ transformation. We use Adam

(Kingma, 2014) for optimization. Additionally, in the 3-agent *MultiHalfCheetah* environment, the tendon structure can destabilize the MuJoCo simulator. To mitigate this, we impose an additional constraint on the output of HATD3-SoCo, clipping it to the range $[-0.85, 0.85]$.

For the PegMARL baseline, its official implementation is designed for discrete action spaces (both discrete-state gridworld and continuous-state MPE environments) and relies on individual reward shaping. As a result, it cannot be directly adapted to off-policy DPG algorithms that depend on a centralized Critic, and it is also not straightforward to apply to one-to-many settings such as *Spread*. Therefore, based on the HARL codebase, we make the following modifications: (i) modify the inputs and outputs of the actor–critic and discriminator to support continuous actions; (ii) following the original paper, use individual critics; and (iii) for *Spread*, where multiple solo views are available, randomly select one solo view as the personal observation. Further discussion of PegMARL can be found in Appendix F.3.

### D.4 HYPERPARAMETERS

Except for the correction strength $L$ in SoCo, all hyperparameters follow the default or recommended (when available) settings in HARL to ensure fair comparison. The detailed configurations are reported in Table 2.

Table 2: Shared hyperparameters for DPG algorithms.

| Hyperparameter | Value | Hyperparameter | Value |
|---|---|---|---|
| Batch Size | 1000 | Buffer Size | 1000000 |
| Hidden Size | 256 (128 for *Spread*) | Discount Factor $\gamma$ | 0.99 |
| $n$-step TD | 10 (1 for *Spread*) | Explore Noise | 0.1 |
| Policy Noise | 0.2 | Noise Clip | 0.5 |
| Policy Delay | 2 | Soft Update Coefficient | 0.005 |
| Actor Learning Rate | 0.0005 | Critic Learning Rate | 0.001 |

For SoCo, $L$ is an important hyperparameter that controls the extent to which knowledge from solo demonstrations is leveraged. The values of $L$ used for each task and backbone algorithm are summarized in Table 3. Different tasks require different $L$ values, as the optimal balance depends on factors such as the degree of domain shift and the inherent difficulty of the cooperative environment.

Table 3: Correction strength $L$ used in SoCo for each task and backbone algorithm.

| Task | MATD3-SoCo | HATD3-SoCo |
|---|---|---|
| Spread-3 | 0 | 0 |
| Spread-4 | 0 | 0 |
| Spread-5 | 0 | 0 |
| LongSwimmer-3 | 3.15 | 2.20 |
| LongSwimmer-4 | 3.10 | 2.90 |
| LongSwimmer-5 | 2.10 | 2.85 |
| MultiHalfCheetah-2 | 1.90 | 2.00 |
| MultiHalfCheetah-3 | 1.90 | 1.90 |
| MultiWalker-2 | 3.00 | 3.00 |

## E ADDITIONAL RESULTS WITH STOCHASTIC POLICY

### E.1 EXTENSION TO HASAC

HASAC (Liu et al., 2024a) is a recently proposed advanced stochastic-policy MARL algorithm that extends SAC to heterogeneous-agent training. When adapting SoCo to this backbone, the main modification lies in how we compute the entropy regularization term $\log \pi(a \mid s)$.

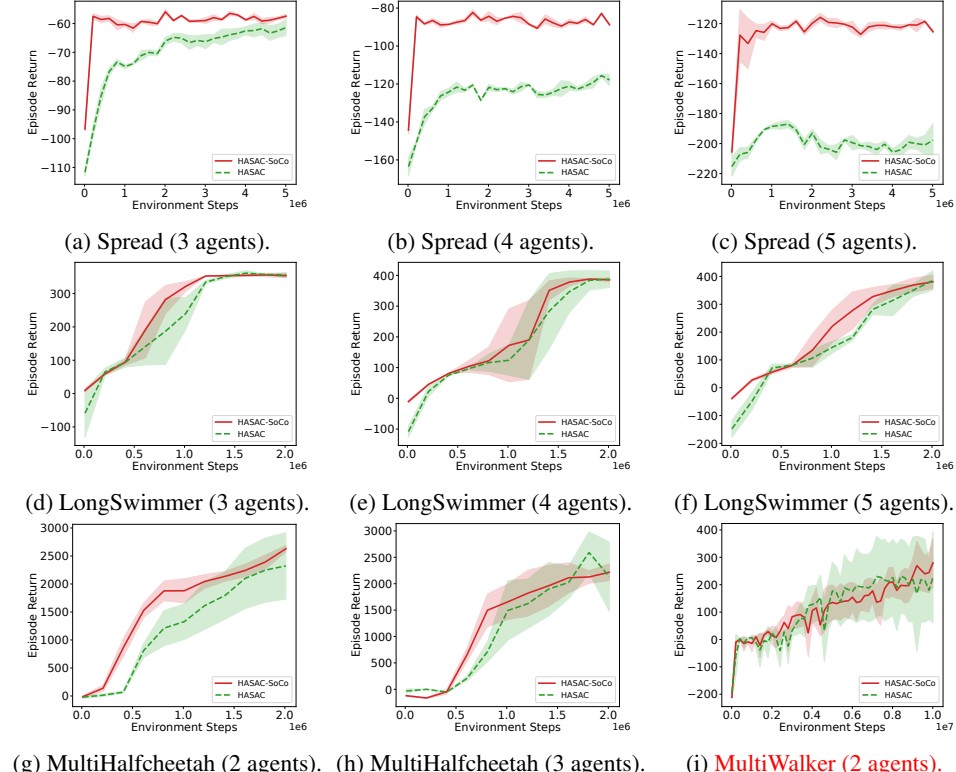

Figure 7: Training curves on nine tasks with HASAC. Results are averaged over three random seeds, with solid and dashed lines indicating the mean performance and shaded areas representing one standard deviation.

Recall that in Section 3.3, we first use the gating weights $\boldsymbol{w} = g_\varphi^i(o_t^i)$ to select an action candidate, and then combine it with the residual adjustment $\Delta a^i$ from the coordination policy $\pi_\theta$ via the action editor. Therefore, we can derive the distribution of the fused action from $\text{GumbelSoftmax}(g_\varphi^i(o_t^i))$ and $\pi_\theta$.

Specifically, we assume that, given any observation, the gate selector and the action editor act independently. Let $\tilde{a}^i \in \{a^{i,1}, \ldots, a^{i,G_i}\}$ denote the candidate action selected by the gate. Then the density of the fused action $a^i$ can be written as

$$\Pi_\phi(a^i \mid o^i) = \sum_{k=1}^{G_i} P(\tilde{a}^i = a^{i,k} \mid o^i) \, P(\Delta a^i = a^i - a^{i,k} \mid o^i).$$

For the squashed policy and the differentiable clipping operator, we just need to introduce the Jacobian correction appropriately.

Given $\Pi_\phi$, we can compute the entropy of the fused action in the standard way. The same construction naturally extends to other stochastic-policy MARL algorithms.

### E.2 EXPERIMENT RESULTS

Similar to the setup in Section 4, we evaluate the performance of SoCo with HASAC on nine tasks across four continuous-control scenarios. All experiments are run with three random seeds, using the hyperparameter configurations recommended by HARL whenever applicable. The values of correction strength $L$ used in SoCo for each task are listed in Table 4. The training results are shown in Figure 7.

The results show that, for a recent advanced stochastic-policy algorithm, SoCo can still consistently accelerate multi-agent training and achieve competitive or even superior final performance compared

to the baseline. This further demonstrates the plug-and-play nature of SoCo and its strong potential to leverage solo demonstrations to improve cooperative training.

Table 4: Correction strength $L$ used in SoCo for each task with HASAC.

| Task | Value | Task | Value | Task | Value |
|------|-------|------|-------|------|-------|
| Spread-3 | 0 | LongSwimmer-3 | 2.6 | MultiHalfCheetah-2 | 2.0 |
| Spread-4 | 0 | LongSwimmer-4 | 2.5 | MultiHalfCheetah-3 | 2.5 |
| Spread-5 | 0 | LongSwimmer-5 | 3.2 | MultiWalker-2 | 3.00 |

## F DISCUSSIONS

### F.1 EXTENSION TO DISCRETE-ACTION ENVIRONMENT

While the idea behind SoCo is indeed insightful, this work only explicitly demonstrates its effectiveness in continuous action spaces. However, the intrinsic characteristics of discrete-action spaces make a direct extension of SoCo non-trivial. The challenge mainly involves two aspects:

**Action Mismatch.** A straightforward extended implementation would apply SoCo at the logits level. However, since discrete actions rely on argmax-based sampling, fine-tuning logits to adjust the final action is extremely difficult. For instance, in our preliminary attempts on the *Protoss 8v8* environment in SMAC-v2 (Ellis et al., 2023), early-stage coordination policy tried to adjust over 50% of actions, yet fewer than 10% of those adjustments successfully changed the executed actions. The mismatch between intended and executed actions limits the exploration, and the alignment only appeared at later training stages.

**Near-saturated Benchmark.** Moreover, the continuous-control MARL tasks we study, though appearing simpler, actually provide more optimization headroom and clearer insight into SoCo's effect on coordination efficiency. In contrast, we have observed that existing MARL algorithms, such as QMIX-style methods and MAPPO, already achieve very high efficiency and performance in most discrete-action benchmarks (e.g., SMAC-v1 (Samvelyan et al., 2019)/-v2 (Ellis et al., 2023), Google Research Football (Kurach et al., 2020)) under implementations like MAPPO's official implementation (Yu et al., 2022), PyMARL2 (Hu et al., 2023), PyMARL3 (Xiaotian et al., 2023) and HARL (Xiaotian et al., 2023). Although these tasks seem to be "more difficult", their efficiency gap compared with SoCo's "intention-matching process" could be minimal, leaving little room for SoCo to bring further improvement. Establishing more complex discrete-action tasks remains an important direction for MARL community.

### F.2 EXTENSION TO GENERAL MARL TASKS

**Unstructured Observation** Although this work assumes that the observation space is structured and decomposable, many practical scenarios involve unstructured observations where it is difficult to manually design decomposition rules. One promising direction for handling such scenarios is to employ LLMs/VLMs as information processors (Cao et al., 2025) to convert raw, unstructured inputs into structured representations suitable for SoCo.

**Non-decomposable Coordination** Another interesting topic is extending SoCo to inherently non-decomposable tasks, for which proxy solo tasks can be designed to capture relevant individual behaviors. For instance, in *MultiWalker* environment, where two walkers must cooperate to lift a heavy object, it is difficult to decompose the cooperative task into solo ones directly. As described in Appendix D, we collect solo demonstrations by constructing a proxy single-walker task in which the agent lifts a lighter object to learn basic standing and lifting behaviors. As shown in Figure 2i, even though these solo demonstrations differ significantly in their dynamics from those of the cooperative task, SoCo still substantially improves the backbone algorithm's training efficiency.

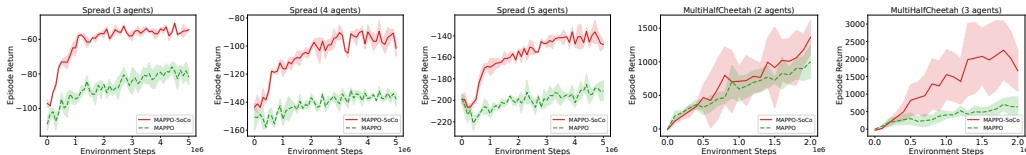

Figure 8: Training curves on five tasks with MAPPO. Results are averaged over three random seeds, with solid and dashed lines indicating the mean performance and shaded areas representing one standard deviation.

**Heteregenous Agents** For heterogeneous agents with different observation or action spaces, SoCo can be extended by incorporating an attention-based mechanism to handle variable-sized inputs (Hu et al., 2021; Zhang et al., 2023a; Liu et al., 2025). For agents with specialized roles, SoCo can train cooperative policies with heterogeneous MARL algorithms, thereby enabling the training of heterogeneous agents. As to the heterogeneity of the solo policy, one possible approach is to leverage techniques from multi-task offline RL, e.g., skill-discovery (Zhang et al., 2023a; Liu et al., 2025), to train role-conditioned solo policies that adapt SoCo's coordination process to heterogeneous settings.

### F.3 COMPARISON WITH PEGMARL

PegMARL (Yu et al., 2025) guides MARL training with personalized demonstratinos, which is highly related to the problem studied in SoCo. However, PegMARL and SoCo are based on fundamentally different assumptions:

PegMARL mainly achieves individual reward reshaping via distribution matching to obtain expert guidance, where its personalized behavior and transition discriminators (Eqs. (9)–(10) in Yu et al. (2025)) require the personalized observation structure to be consistent with that of the cooperative environment. Although, when the cooperative environment involves multiple agents, like SoCo, PegMARL uses a decomposer to extract observations that are compatible with the single-agent setting (e.g., in the MultiHalfCheetah scenario), it still cannot handle multiple solo views (e.g., in the Spread scenario), whereas SoCo resolves this issue via a learnable gating selector.

Moreover, the individual reward shaping mechanism of PegMARL makes it difficult to directly integrate with standard CTDE algorithms that rely on a centralized critic and a shared team reward (e.g., MADDPG (Lowe et al., 2017), MATD3 (Ackermann et al., 2019), HATD3 (Zhong et al., 2024), HASAC (Liu et al., 2024a)), and it is instead more naturally suited to decentralized methods (e.g., Independent PPO (Yu et al., 2022)). By contrast, SoCo is primarily designed for CTDE algorithms. Thus, the two should be regarded as orthogonal techniques, and a direct comparison between PegMARL and SoCo under our setting is not entirely appropriate. Nevertheless, we believe PegMARL and SoCo are compatible rather than conflicting, and exploring how to combine them on certain tasks is an interesting direction for future work.

### F.4 APPLICABILITY OF SOCO TO ON-POLICY METHODS

While SoCo can effectively improve the performance of off-policy MARL algorithms through the gating selector and action editor, our experiments indicate that directly combining SoCo with less sample-efficient on-policy methods such as MAPPO can be challenging. To illustrate these challenges, we evaluate MAPPO and its SoCo variants on five tasks across the Spread and Multi-HalfCheetah scenarios in Figure 8. We find that, although the gating mechanism still performs well on Spread, directly applying the action editor in the MultiHalfCheetah tasks is ineffective. In particular, because MAPPO relies on clipped importance ratios to constrain the magnitude of each policy update, merely tuning the correction strength $L$ is insufficient for SoCo to quickly adapt the solo policy to a new cooperative environment; alleviating this issue requires enlarging the clipping range (e.g., tuning clipping parameter $\epsilon$ to $0.5$). Moreover, the lower sample efficiency of on-policy methods further hinders rapid policy transfer in some settings. Therefore, designing more suitable ways to exploit solo demonstrations specifically for on-policy algorithms is an important direction for future work.

