# OpenReview forum: "From Solo to Symphony: Orchestrating Multi-Agent Collaboration with Single-Agent Demos"
_ICLR.cc/2026/Conference — Submitted to ICLR 2026_

### Official Review · Reviewer_wccp · 2025-10-19

**Soundness:** 2
**Presentation:** 2
**Contribution:** 2
**Rating:** 4
**Confidence:** 3

**Summary:**

The paper introduces "Solo-to-Collaborative RL (SoCo)", a framework designed to leverage single-agent (solo) demonstrations to accelerate multi-agent reinforcement learning (MARL) in cooperative settings. The authors argue that while MARL typically requires expensive multi-agent data, solo demonstrations are often more abundant and easier to obtain (e.g., in robotics or coding tasks). SoCo addresses challenges like observation mismatch and domain shift by: (1) pretraining a shared solo policy via behavior cloning on solo demos; (2) decomposing multi-agent observations into solo-compatible views during training; and (3) using a policy fusion module with a learnable gating selector (to choose among candidate solo actions) and an action editor (to refine actions via residual corrections). The framework is plug-and-play with CTDE-based MARL algorithms like MATD3 and HATD3. Experiments on nine tasks across four benchmarks (Spread, LongSwimmer, MultiHalfCheetah, MultiWalker) demonstrate improved sample efficiency and final performance.

**Strengths:**

The framework is a fresh angle compared to prior work focused on multi-agent data (e.g., offline MARL or multi-task transfer).

Results show consistent efficiency gains  and performance boosts, with ablations isolating component contributions (e.g., gating selector vs. random alternatives).

**Weaknesses:**

**Assumptions and Generalizability:** The framework assumes decomposable, structured observations, which may not hold in more complex or unstructured environments. It also relies on rule-based decomposition, limiting applicability to tasks without clear solo-multi alignments. Extension to stochastic policies or non-CTDE paradigms is mentioned but not demonstrated.

**Limited Scope of Baselines and Tasks:** While MATD3/HATD3 are solid, comparisons to other CTDE algorithms (e.g., QMIX, MAPPO) or recent transferable MARL   would strengthen claims. Using real-world offline data would help as well.

 **Theoretical Analysis:** No formal analysis of convergence or transfer guarantees, which    would add depth.

**Questions:**

The correction strength L is task-specific and requires tuning, potentially undermining plug-and-play claims. Ablations show sensitivity (e.g., small L over-relies on solos, large L underuses them), is it able to do the auto-tuning for it.

---

> ### Author Response · Authors · 2025-11-20
> **Response to Reviewer wccp (Part 1/2)**
>
> Thank you for your time and effort in reviewing our paper! We are grateful for your constructive suggestions, which have significantly guided our improvements. Please find our responses to your comments below.
>
> ### Weaknesses
> > **Weakness1:** Assumptions and Generalizability: The framework assumes decomposable, structured observations, which may not hold in more complex or unstructured environments. It also relies on rule-based decomposition, limiting applicability to tasks without clear solo-multi alignments. Extension to stochastic policies or non-CTDE paradigms is mentioned but not demonstrated.
>
> Thank you for your comment! We would like to emphasize that SoCo reveals the **potential of leveraging solo demonstrations from non-cooperative environments** to facilitate cooperative MARL training. While the assumption of *well-defined, structured, and decomposable* tasks is common in multi-task (offline) MARL (e.g., [1],[2], and [3]), it is **reasonable for a wide range of real-world problems**, such as **network scheduling, code generation, and navigation**, where structured observations and decomposable subtasks naturally exist.
>
> For non-decomposable tasks, **proxy solo tasks** can be designed to capture relevant individual behaviors. For instance, in our **MultiWalker** environment, two walkers must cooperate to lift a heavy object. As described in **Appendix D**, we collect solo demonstrations by constructing a proxy single-walker task in which the agent lifts a lighter object to learn basic standing and lifting behaviors. As shown in **Figure 2(i)**, even though these solo demonstrations differ significantly in dynamics from the cooperative task, **SoCo still substantially improves the backbone algorithm’s training efficiency**.
>
> Regarding unstructured observations and the rule-based decomposition, one promising direction is to employ **LLMs/VLMs as information processors**[4] to convert raw, unstructured inputs into structured representations suitable for SoCo. However, this extension is **beyond the scope** of the current work.
>
> We have added these discussions in Appendix F.2 in our revision.
>
> As to potential extension, we would like to clarify that our work **does not involve non-CTDE paradigms**. On the contrary, **SoCo fully follows the CTDE framework**. Regarding **stochastic policies**, we have **added additional experiments** in Appendix E using **HASAC** as stochastic-policy backbones. The **consistent improvements** achieved by SoCo over this recent advanced algorithm further validate the generality of our approach.
>
> #### **References**
> ```
> [1] Hu, S. et al., UPDeT: Universal Multi-agent RL via Policy Decoupling with Transformers. ICLR 2021.
> [2] Zhang, F. et al., Discovering Generalizable Multi-agent Coordination Skills from Multi-task Offline Data. ICLR 2023.
> [3] Liu, S. et al., Learning Generalizable Skills from Offline Multi-Task Data for Multi-Agent Cooperation. ICLR 2025.
> [4] Cao, Y., Survey on Large Language Model-Enhanced Reinforcement Learning: Concept, Taxonomy, and Methods. TNNLS 2025.
> ```
>
> > **Weakness2:** Limited Scope of Baselines and Tasks: While MATD3/HATD3 are solid, comparisons to other CTDE algorithms (e.g., QMIX, MAPPO) or recent transferable MARL would strengthen claims. Using real-world offline data would help as well.
>
> Thank you for your suggestion! We have added additional experiments in Appendix E based on HASAC[1], **a recently proposed and advanced stochastic-policy** MARL algorithm.
>
> The results show that, for a recent advanced stochastic-policy algorithm, SoCo can still **consistently accelerate multi-agent training and achieve competitive or even superior final performance** compared to the baseline. This further demonstrates the plug-and-play nature of SoCo and its strong potential to leverage solo demonstrations to improve cooperative training.
>
>
> Additionally, we would like to emphasize that **SoCo explores a new and open problem**: whether **offline solo demonstrations** can accelerate **cooperative MARL** training. This setting is **fundamentally different** from standard **transferable MARL**. In such *solo-to-multi* scenario, the algorithm has access only to **solo demonstrations pre-collected from non-cooperative environments** during the offline phase, and these observations often **mismatch** those in the cooperative environment. To the best of our knowledge, there are **no existing baselines** that can be directly applied to this setting.
>
> We agree that valuating on real-world datasets would provide stronger practical validation, and we plan to explore this direction in future work.
>
> #### **Reference**
> ```
> [1] Liu, J. et al., Maximum Entropy Heterogeneous-Agent Reinforcement Learning. ICLR 2024.
> ```

---

> > ### Author Response · Authors · 2025-11-20
> > **Response to Reviewer wccp (Part 2/2)**
> >
> > > **Weakness3:** Theoretical Analysis: No formal analysis of convergence or transfer guarantees, which would add depth.
> >
> > Thank you for your suggestion! However, existing theoretical work on multi-agent learning, e.g., [1] and [2], is typically developed under **highly structured assumptions** (such as finite state–action spaces, non–deep function approximators, and oracle access), which **differ substantially from our deep neural network–based solo-to-multi transfer setting**. As a result, providing a formal theoretical analysis with convergence or transfer guarantees in our setting remains **largely an open problem**, and we view this as an important direction for future work.
> >
> > #### **References**
> > ```
> > [1] Zhang, Q. et al., Near-Optimal Online Learning for Multi-Agent Submodular Coordination: Tight Approximation and Communication Efficiency. ICLR 2025.
> > [2] Ocello, A. et al., Finite-Sample Convergence Bounds for Trust Region Policy Optimization in Mean Field Games. ICML 2025
> > ```
> >
> > ### Questions
> > > The correction strength L is task-specific and requires tuning, potentially undermining plug-and-play claims. Ablations show sensitivity (e.g., small L over-relies on solos, large L underuses them), is it able to do the auto-tuning for it.
> >
> > Thank you for your question. The correction strength $L$ is indeed an important hyperparameter in SoCo. How to automatically adjust it while ensuring policy stability is an interesting direction for future research. However, we believe this **does not undermine the plug-and-play nature** of SoCo. On the contrary, $L$ serves as a tunable knob that allows SoCo to **adapt flexibly across different downstream tasks**, offering greater generality and control rather than added complexity.
> >
> > ---
> > We hope our response addresses your concerns. If so, we wonder if you could kindly consider raising your score? We will also be happy to answer any further questions you may have. Thank you very much!

---

> ### Author Response · Authors · 2025-11-25
> **Looking Forward to Your Feedback**
>
> Dear Reviewer wccp,
>
> Thank you for your time and feedback! We hope our response and the revision have fully addressed your concerns. If so, we would appreciate it if you could raise your score.
>
> If you have any remaining concerns, we are more than happy to discuss them further.
>
> Best regards,
>
> The Authors

---

> > ### Comment · Reviewer_wccp · 2025-11-26
> >
> > Including more tasks and baselines, e.g., MAPPO and PegMARL, in experiments will be better.
> >
> > Currently, I keep my score.

---

> ### Author Response · Authors · 2025-12-02
> **Response to the Official Comment by Reviewer wccp**
>
> Thank you very much for the suggestion! We have followed your advice and added MAPPO and PegMARL as additional baselines in Figure 2. The new results show that MATD3-SoCo and HATD3-SoCo still **achieve comparable or even superior performance against both MAPPO and PegMARL across our tasks**.
>
> We would like to note, however, that these comparisons are not entirely apples-to-apples. MAPPO is an *on-policy* method, and PegMARL is more suitable for *fully decentralized* paradigm, while our main focus is on *off-policy CTDE* algorithms. We therefore adapt and port both methods to our scenario, mainly to demonstrate that **SoCo not only improves over its backbones, but also achieves competitive (and often superior) performance compared to representative MARL algorithms under this setting**.
>
> In addition, we have added a new discussion in Appendix F.4 on the applicability of SoCo to on-policy backbones. Empirically, we find that, due to the lower sample efficiency of on-policy methods and the constraints imposed by their policy-based updates, SoCo can be directly effective on some tasks but requires modifying the backbone algorithm itself to work well on others. Designing more effective ways to exploit solo demonstrations for on-policy algorithms is therefore an interesting and important direction for future work.

---

### Official Review · Reviewer_fw46 · 2025-10-29

**Soundness:** 2
**Presentation:** 2
**Contribution:** 3
**Rating:** 2
**Confidence:** 3

**Summary:**

This paper introduces Solo-to-Cooperative Reinforcement Learning (SoCo), a framework designed to leverage solo demonstrations to improve the efficiency of multi-agent reinforcement learning (MARL). SoCo first pretrains a shared solo policy from individual-agent data and then adapts it for cooperation through a policy fusion mechanism consisting of a gating selector and an action editor. Experimental results across multiple cooperative benchmarks demonstrate that SoCo accelerates convergence and achieves competitive or superior performance, showing that solo experiences can serve as a scalable and effective complement to multi-agent data.

**Strengths:**

1. The paper tackles a practical and underexplored challenge in MARL - how to utilize single-agent (solo) demonstrations to enhance multi-agent cooperation. This "solo-to-cooperative" perspective is conceptually novel and practically valuable for domains where cooperative data are costly or difficult to obtain.
2. The proposed SoCo framework is modular and algorithm-agnostic, allowing seamless integration with existing MARL methods (e.g., MATD3). This design highlights strong flexibility and reusability, making SoCo a potentially general framework for future MARL research.

**Weaknesses:**

1. The experimental environments used in the paper are not fully representative of mainstream MARL benchmarks. Evaluating SoCo on widely recognized cooperative environments such as SMAC or MaMuJoCo would significantly strengthen the empirical evidence and better demonstrate its capability in handling complex coordination scenarios.
2. The paper lacks comparisons with recent advances in transferable reinforcement learning and multi-agent imitation learning methods. Without such baselines, it is difficult to judge the generality and competitiveness of SoCo relative to the state of the art. Incorporating methods from the past year would make the empirical study more comprehensive and convincing.
3. The notations in Figure 1 are inconsistent with those in the main text, in particular, the figure uses subscripts starting from 1, while the text uses k = 0. Unifying the notation and indexing conventions would improve clarity and technical precision.

**Questions:**

1. Could the authors provide a more concrete or simplified explanation of the observation decomposition process? A small illustrative example would greatly help clarify how observations are partitioned into solo views and used in the SoCo framework.
2. The authors claim that in the 2-agent MultiHalfCheetah environment, the absence of multi-objective tasks allows for an "effective isolation of the gating selector's influence."Under this environment, does the decomposition process still generate multiple solo views? Please clarify this point.If the gating selector is not truly isolated, please explain the following:When L = 0, the MultiHalfCheetah experiment yields near-zero returns, indicating that direct transfer of solo policies fails due to domain shift. However, in the 3-agent Spread ablation study with the same setting (L = 0), the performance remains high. These two results seem inconsistent - could the authors clarify the underlying reason for this discrepancy?

---

> ### Author Response · Authors · 2025-11-20
> **Response to Reviewer fw46 (Part 1/3)**
>
> Thank you for your time and effort in reviewing our paper! We are grateful for your constructive suggestions, which have significantly guided our improvements. Please find our responses to your comments below.
>
> ### Weaknesses
> > **Weakness1:** The experimental environments used in the paper are not fully representative of mainstream MARL benchmarks. Evaluating SoCo on widely recognized cooperative environments such as SMAC or MaMuJoCo would significantly strengthen the empirical evidence and better demonstrate its capability in handling complex coordination scenarios.
>
> Thank you for your suggestion! We would like to clarify that our work primarily focuses on **continuous-action** MARL, and all evaluation environments are drawn from **mainstream MARL benchmarks for continuous-action**, including *Spread* (from **MPE**), *LongSwimmer* and *MultiHalfCheetah* (from **MAMuJoCo**), and *MultiWalker* (from **SISL**). Our experiments show that these tasks, particularly Spread-5 and MultiWalker, remain **highly challenging** for existing MARL baselines, as reflected in their training curves.
>
> In contrast, SMAC and related environments mainly involve **discrete action spaces**. As discussed in the revision, **extending SoCo to discrete-action scenarios is non-trivial, and we regard it as an important future direction**. We have explicitly added a discussion on this limitation and potential extensions in the revised manuscript.
>
> Specifically, we would like to share several insights into the challenges involved:
>
> - A straightforward implementation would apply SoCo at the logits level. However, since discrete actions rely on **argmax-based sampling**, fine-tuning logits to adjust the final action is extremely difficult. For instance, in our preliminary attempts on the *SMAC-v2 Protoss 8v8* environment, early-stage coordination policy tried to adjust over 50% of actions, yet fewer than 10% of those adjustments successfully changed the executed actions. The mismatch between intended and executed actions limits the exploration, and the alignment only appeared at later training stages.
>
> - Moreover, the continuous-control MARL tasks we study, though appearing simpler, **actually provide more optimization headroom and clearer insight into SoCo’s effect on coordination efficiency.** In contrast, we have observed that existing MARL algorithms, such as QMIX-style methods and MAPPO, already achieve very high efficiency and performance in most discrete-action benchmarks (e.g., SMAC-v1/-v2, Google Research Football) under implementations like MAPPO's official implementation[1], PyMARL2[2], PyMARL3[3] and HARL[4]. Although these tasks seem to be "more difficult", their efficiency gap compared with SoCo’s “intention-matching process” could be minimal, leaving little room for SoCo to bring further improvement. **Establishing more complex discrete-action tasks remains an important direction for MARL community.**
>
> We have added the related discussions in Appendix F.1.
>
> #### **References**
> ```
> [1] Yu, C., The Surprising Effectiveness of PPO in Cooperative Multi-Agent Games. NeurIPS 2022.
> [2] Hu, J. et al., Rethinking the Implementation Tricks and Monotonicity Constraint in Cooperative Multi-Agent Reinforcement Learning. ArXiv 2021, ICLR 2023 Blog Track.
> [3] Hao, X. et al., Boosting Multi-Agent Reinforcement Learning via Permutation Invariant and Permutation Equivariant Networks. ICLR 2023.
> [4] Zhong, Y. et al., Heterogeneous-agent reinforcement learning. JMLR 2024.
> ```

---

> > ### Author Response · Authors · 2025-11-20
> > **Response to Reviewer fw46 (Part 2/3)**
> >
> > > **Weakness2:** The paper lacks comparisons with recent advances in transferable reinforcement learning and multi-agent imitation learning methods. Without such baselines, it is difficult to judge the generality and competitiveness of SoCo relative to the state of the art. Incorporating methods from the past year would make the empirical study more comprehensive and convincing.
> >
> > Thank you for your suggestion! We would like to emphasize that **SoCo explores a new and open problem**: whether **offline solo demonstrations** can accelerate **cooperative MARL** training. This setting is **fundamentally different** from standard **transfer reinforcement learning** or **multi-agent imitation learning**. In such *solo-to-multi* scenario, the algorithm has access only to **solo demonstrations pre-collected from non-cooperative environments** during the offline phase, and these observations often **mismatch** those in the cooperative environment. To the best of our knowledge, there are **no existing baselines** that can be directly applied to this setting.
> >
> > Nevertheless, our experiments cover a **diverse set of cooperative tasks and backbone algorithms**, and we also add an additional experiment based on HASAC[1], **a recently proposed and advanced stochastic-policy MARL algorithm**, in Appendix E.
> >
> > The results show that, for a recent advanced stochastic-policy algorithm, SoCo can still **consistently accelerate multi-agent training and achieve competitive or even superior final performance** compared to the baseline. This further demonstrates the plug-and-play nature of SoCo and its strong potential to leverage solo demonstrations to improve cooperative training, highlighting the **novelty and pioneering contribution** of this work.
> >
> >
> > #### **Reference**
> > ```
> > [1] Liu, J. et al., Maximum Entropy Heterogeneous-Agent Reinforcement Learning. ICLR 2024.
> > ```
> >
> > > **Weakness3:** The notations in Figure 1 are inconsistent with those in the main text, in particular, the figure uses subscripts starting from 1, while the text uses k = 0. Unifying the notation and indexing conventions would improve clarity and technical precision.
> >
> > Thank you for your suggestion! We have revised these notation issues; the related subscripts should start from 1.
> >
> > ### Questions
> > > **Question1:** Could the authors provide a more concrete or simplified explanation of the observation decomposition process? A small illustrative example would greatly help clarify how observations are partitioned into solo views and used in the SoCo framework.
> >
> > Thank you for your question. We use the **3-agent Spread** task as an example to illustrate the observation decomposition process.
> >
> > - In the solo demonstration, there is only one agent and one landmark, so the agent’s observation consists of [`own_pos`, `own_vel`, `landmark_pos_r`], where these components respectively denote the agent’s position, velocity, and the relative position of the landmark.
> >
> > - In the 3-agent cooperative environment, each agent additionally observes other agents and landmarks: [`own_pos`, `own_vel`, `pos_r_1`, `pos_r_2`, `landmark_pos_r_1`, `landmark_pos_r_2`, `landmark_pos_r_3`, `comm_1`, `comm_2`], where `pos_r_i` is the position of the $i$-th agent relative to itself, `landmark_pos_r_i` is the relative position of the $i$-th landmark, and `comm_i` represents the communication message from agent $i$ (set to 0 in our experiments as communication is disabled).
> >
> > - We then **decompose each local observation** into three *solo views*:
> >
> >     - [`own_pos`, `own_vel`, `landmark_pos_r_1`]
> >     - [`own_pos`, `own_vel`, `landmark_pos_r_2`]
> >     - [`own_pos`, `own_vel`, `landmark_pos_r_3`]
> >
> > each of which is passed to the **solo policy** to generate corresponding action candidates. This illustrative example has been added to Appendix D.2.3 for clarity.

---

> > > ### Author Response · Authors · 2025-11-20
> > > **Response to Reviewer fw46 (Part 3/3)**
> > >
> > > > **Question2:** The authors claim that in the 2-agent MultiHalfCheetah environment, the absence of multi-objective tasks allows for an "effective isolation of the gating selector's influence."Under this environment, does the decomposition process still generate multiple solo views? Please clarify this point.If the gating selector is not truly isolated, please explain the following:When L = 0, the MultiHalfCheetah experiment yields near-zero returns, indicating that direct transfer of solo policies fails due to domain shift. However, in the 3-agent Spread ablation study with the same setting (L = 0), the performance remains high. These two results seem inconsistent - could the authors clarify the underlying reason for this discrepancy?
> > >
> > > Thank you for your question! We would like to clarify that in the **2-agent MultiHalfCheetah** environment, there are **no multiple solo views**, since each local observation corresponds only to the agent's own state and other agents' state. Thus, there is **no one-to-many correspondence** between the cooperative and solo tasks (i.e., MultiHalfCheetah vs. HalfCheetah). Consequently, when $L = 0$, the change in environment dynamics causes the direct transfer of the solo policy to completely fail, leading to near-zero returns.
> > >
> > > In contrast, in the **Spread** task, each agent observes multiple landmarks, creating a **one-to-many mapping** between the cooperative and solo settings (i.e., 3-agent Spread vs. 1-agent Spread). Moreover, the **dynamics between the two environments are almost identical**, so even with $L = 0$, the learnable gating network can **select the appropriate target landmark step-wise**, effectively solving the navigation task.
> > >
> > > ---
> > > We hope our response addresses your concerns. If so, we wonder if you could kindly consider raising your score? We will also be happy to answer any further questions you may have. Thank you very much!

---

> ### Author Response · Authors · 2025-11-25
> **Looking Forward to Your Feedback**
>
> Dear Reviewer fw46,
>
> Thank you for your time and feedback! We hope our response and the revision have fully addressed your concerns. If so, we would appreciate it if you could raise your score.
>
> If you have any remaining concerns, we are more than happy to discuss them further.
>
> Best regards,
>
> The Authors

---

> > ### Comment · Reviewer_fw46 · 2025-11-25
> >
> > Thanks a lot for your detailed reply, which I think has addressed most of my concerns. I have adjusted my rating accordingly.

---

> ### Author Response · Authors · 2025-11-25
> **Response to Official Comment by Reviewer fw46**
>
> Thank you very much for your response and for raising your score! If there are any remaining concerns we could clarify, we would be happy to address them. If not, we would be very grateful if you could consider increasing your score to a positive recommendation.

---

### Official Review · Reviewer_M9Em · 2025-11-07

**Soundness:** 3
**Presentation:** 3
**Contribution:** 2
**Rating:** 2
**Confidence:** 5

**Summary:**

This paper presents Solo-to-Collaborative RL (SoCo), a framework designed to leverage single-agent (“solo”) demonstrations to accelerate cooperative multi-agent reinforcement learning. The premise is well-motivated.. it is often easier to obtain solo demos than multi-agent demos. However, solo demos do not show any cooperation even if they do have signals on how each agent should act. The key technical challenge is how to determine when to mimic the solo demos and when to learn to cooperate. The method first trains a shared solo policy via imitation learning, then during multi-agent training decomposes each agent’s local observation into multiple solo-compatible views and reuses the pretrained policy to generate candidate actions. These candidates are fused through a learnable module consisting of a gating selector (for resolving ambiguity and selecting a solo action) and an action editor (which applies residual corrections to adapt to cooperative dynamics).

The method is evaluated in simulation comparing two MARL algorithms with and without SoCo. The results show improved speed of convergence and better convergence with the SoCo. Overall, this is a well written paper and explores and well-motivated idea. However, I have reservations about the limited empirical evaluation and no comparisons with other methods (including PegMARL) that address a similar problem.

**Strengths:**

- The empirical results clearly show improvement with the SoCo framework on top of two existing MARL algorithms.
- The paper addresses a well motivated and under-studied problem and the approach seems sound.
- The idea of action editor is interesting and perhaps can find applications beyond the solo-to-multi setting addressed in this paper. The key challenged addressed in this paper is domain shift and the action editor idea can be used in other settings where domain shift can be observed.

**Weaknesses:**

My main reservation is the lack of comparisons with other baseline methods and limited evaluation. For example, the paper claims that PegMARL (Yu et al., 2025) "assume sufficient multi-agent data" but that is not true. Most of the experiments in the PegMARL paper are with single agent demonstrations (see results in Sec 5.1). In fact, it seems that PegMARL would be a more general version since it can handle both single agent as well as multi-agent demonstrations (Sec 5.2). Given that, it seems that SoCo should be benchmarked against PegMARL.

The empirical evaluations also only consider two MARL algorithms (with a similar flavor) and compare those with and without SoCo. It would seem appropriate to benchmark with other MARL algorithms (e.g., MAPPO) and other simpler baseline algorithms (naively using single agent demos to bootstrap learning multi-agent policies).

The paper also seems to focus on cases where the agents are homogenous. It would be interesting to discuss whether the framework extends to heteregenous cases where agents may have specialized capabilities or roles to play.

**Questions:**

- How does SoCo compare with other methods solving similar problems including PegMARL?

- How does SoCo compare with other MARL algorithms such as MAPPO?

- How easy/difficult is it to disentangle joint observations into solo ones? It seems like this is done by hand, if so, how can this generalize to other MARL settings?

- How does the quality of the demonstrations affect the performance?

- Can you comment on whether SoCo extends to heterogeneous agents with different observation spaces, action spaces, or specialized roles?

---

> ### Author Response · Authors · 2025-11-20
> **Response to Reviewer M9Em (Part 1/3)**
>
> Thank you for your time and effort in reviewing our paper! We are grateful for your constructive suggestions, which have significantly guided our improvements. Please find our responses to your comments below.
>
> ### Weaknesses
> > **Weakness1:** My main reservation is the lack of comparisons with other baseline methods and limited evaluation. For example, the paper claims that PegMARL (Yu et al., 2025) "assume sufficient multi-agent data" but that is not true. Most of the experiments in the PegMARL paper are with single agent demonstrations (see results in Sec 5.1). In fact, it seems that PegMARL would be a more general version since it can handle both single agent as well as multi-agent demonstrations (Sec 5.2). Given that, it seems that SoCo should be benchmarked against PegMARL.
>
> Thank you for your detailed comment and suggestion! However, we would like to clarify that **PegMARL's demonstrations not a more general version of SoCo's.**
>
> Instead, PegMARL and SoCo are based on fundamentally different assumptions, making direct comparison not straightforward:
>
> - PegMARL achieves expert guidance mainly via per-agent reward reshaping through distribution matching. According to its paper and code, it is best suited to *fully decentralized algorithms with discrete actions*. While PegMARL can be adapted to handle multiple agents in the observation by decomposing it as in SoCo, it still **struggles to handle observations with multiple solo views**.
>
> - In contrast, SoCo is designed for *continuous-action CTDE* algorithms and uses a learnable gating selector to handle multiple solo views.
>
> We have corrected our earlier description of PegMARL and added a more detailed discussion of this distinction in Appendix F.3. Nevertheless, we still adapt PegMARL as a cross-method baseline, and the results show that **SoCo significantly outperforms PegMARL under our setting**.
>
> > **Weakness2:** The empirical evaluations also only consider two MARL algorithms (with a similar flavor) and compare those with and without SoCo. It would seem appropriate to benchmark with other MARL algorithms (e.g., MAPPO) and other simpler baseline algorithms (naively using single agent demos to bootstrap learning multi-agent policies).
>
> Thank you for your suggestion! We have added additional experiments in Appendix E based on HASAC[1], **a recently proposed and advanced stochastic-policy** MARL algorithm, which is different from MATD3 and HATD3. The results show that, for a recent advanced stochastic-policy algorithm, SoCo can still **consistently accelerate multi-agent training and achieve competitive or even superior final performance** compared to the baseline.
>
> Regarding MAPPO, SoCo is primarily built on sample-efficient off-policy algorithms, whereas MAPPO is an on-policy method, so a direct comparison is not strictly fair. Nevertheless, we include MAPPO as a cross-method baseline and compare it with SoCo variants. The results show that **SoCo significantly outperforms MAPPO on almost all tasks**, with the only exception that MAPPO slightly surpasses MATD3-SoCo on MultiWalker at 8M steps.
>
> #### **Reference**
> ```
> [1] Liu, J. et al., Maximum Entropy Heterogeneous-Agent Reinforcement Learning. ICLR 2024.
> ```
>
> > **Weakness3:** The paper also seems to focus on cases where the agents are homogenous. It would be interesting to discuss whether the framework extends to heteregenous cases where agents may have specialized capabilities or roles to play.
>
> Thank you for your suggestion! We would like to emphasize that, as a plug-and-play framework, SoCo can **train cooperative policies with heterogeneous MARL algorithms (e.g., HARL[1])**, thereby enabling the training of heterogeneous agents.
>
> If you are referring to the heterogeneity of the *solo policy*, extending it to heterogeneous agents is indeed an interesting and valuable direction, which may relate to **multi-task offline learning**. One possible approach is to introduce a **skill/role discovery module** before the gating network (e.g., [2], [3], and [4]), and then train a **skill- or role-conditioned gating and policy network**. We plan to explore this extension in future work.
>
> We have added these discussions in Appendix F.2.
> #### **References**
> ```
> [1] Zhong, Y. et al., Heterogeneous-agent reinforcement learning. JMLR 2024.
> [2] Wang, T. et al., RODE: Learning Roles to Decompose Multi-Agent Tasks. ICLR 2021.
> [3] Zhang, F. et al., Discovering Generalizable Multi-agent Coordination Skills from Multi-task Offline Data. ICLR 2023.
> [4] Liu, S. et al., Learning Generalizable Skills from Offline Multi-Task Data for Multi-Agent Cooperation. ICLR 2025.
> ```

---

> ### Author Response · Authors · 2025-11-20
> **Response to Reviewer M9Em (Part 2/3)**
>
> ### Questions
>
> > **Question1:** How does SoCo compare with other methods solving similar problems including PegMARL?
>
> Thank you for your question! We study a new and largely unexplored problem: **how to leverage solo demonstrations to improve MARL training efficiency**, for which there are **very few** existing baseline methods. The cross-method evaluation results show that **SoCo significantly outperforms PegMARL under our setting**.
>
> Nevertheless, we emphasize that while PegMARL is related, as mentioned in our response to Weakness 1, SoCo and PegMARL are based on **fundamentally different assumptions**, making a direct comparison between the two methods difficult.
>
> > **Question2:** How does SoCo compare with other MARL algorithms such as MAPPO?
>
> Thank you for your question! As mentioned in weakness 2, SoCo is primarily built on sample-efficient *off-policy* algorithms, whereas MAPPO is an *on-policy* method, so **a direct comparison is not strictly fair**. Nevertheless, we include MAPPO as a cross-method baseline and compare it with SoCo variants. The results show that **SoCo significantly outperforms MAPPO on almost all tasks**, with the only exception that MAPPO slightly surpasses MATD3-SoCo on MultiWalker at 8M steps.
>
> We also add additional experiments in Appendix E based on HASAC[1], a recently proposed and advanced *off-policy stochastic-policy* MARL algorithm, which is different from MATD3 and HATD3. The results show that, for a recent advanced stochastic-policy algorithm, SoCo can **still consistently accelerate multi-agent training and achieve competitive or even superior final performance compared to the baseline.**
>
> #### **Reference**
> ```
> [1] Liu, J. et al., Maximum Entropy Heterogeneous-Agent Reinforcement Learning. ICLR 2024.
> ```
>
> > **Question3:** How easy/difficult is it to disentangle joint observations into solo ones? It seems like this is done by hand, if so, how can this generalize to other MARL settings?
>
> Thank you for your question! In our setting, the **observations are structurally organized and decomposable**, and we assume that the **decomposition rule is available as domain knowledge**, making this process relatively straightforward. However, we agree that in more **general MARL settings**, learning how to automatically disentangle joint observations into solo ones is an **interesting and valuable direction**. One possible approach is to leverage **LLMs/VLMs as information processors** [1] to transform unstructured joint observations into structured, decomposable representations.
>
> We have added related discussions in Appendix F.2 in the revision.
>
> #### **Reference**
> ```
> [1] Cao, Y., Survey on Large Language Model-Enhanced Reinforcement Learning: Concept, Taxonomy, and Methods. TNNLS 2025.
> ```
>
> > **Question4:** How does the quality of the demonstrations affect the performance?
>
> Thank you for your question! In our setting, high-quality solo demonstrations provide a strong foundation for cooperative training. Consequently, when the demonstration quality decreases, the improvement brought by SoCo during the online joint training phase will diminish, and the framework will need to rely more heavily on the coordination policy to correct suboptimal behaviors.
>
> However, in the additional experiments we included in Section 4.4, we observed several **interesting phenomena** when varying the demonstration quality: the expert policy in the solo environment is **not always** the most beneficial for the cooperative task.
>
> Specifically, we pretrained the solo policy using *poor*, *medium*, and *expert* demonstrations and evaluated them in the MultiHalfCheetah-2 cooperative task:
>
> - With *poor* demonstrations, the solo agent **fails to run stably**, resulting in **slower training and lower final performance**.
>
> - With *medium* demonstrations, the solo agent **learns stable but slow locomotion**, leading to **slower initial progress** than the expert policy. Surprisingly, this smoother control pattern **matches** the dynamics of the cooperative task (where each agent is effectively lighter), yielding higher **final performance** than when pretrained with expert demonstrations.
>
> These results highlight a subtle relationship between solo demonstrations and downstream cooperative training. This also raises an interesting open question: **how should one design solo demonstrations that are most aligned with the downstream multi-agent objective?** We plan to explore this direction in future work.

---

> > ### Author Response · Authors · 2025-11-20
> > **Response to Reviewer M9Em (Part 3/3)**
> >
> > > **Question5:** Can you comment on whether SoCo extends to heterogeneous agents with different observation spaces, action spaces, or specialized roles?
> >
> > Thank you for your question! For **heterogeneous agents with different observation or action spaces**, SoCo can be extended by incorporating an **attention-based mechanism** to handle variable-sized inputs (e.g., [1], [2], and [3]).
> >
> > For agents with **specialized roles**, as discussed in **Weakness 3**, SoCo can **train cooperative policies with heterogeneous MARL algorithms**, thereby enabling the training of heterogeneous agents. If you are referring to the heterogeneity of the *solo policy*, one possible approach is to leverage techniques from **multi-task offline RL** to train **role-conditioned solo policies** that adapt SoCo’s coordination process to heterogeneous settings.
> >
> > #### **References**
> > ```
> > [1] Hu, S. et al., UPDeT: Universal Multi-agent RL via Policy Decoupling with Transformers. ICLR 2021.
> > [2] Zhang, F. et al., Discovering Generalizable Multi-agent Coordination Skills from Multi-task Offline Data. ICLR 2023.
> > [3] Liu, S. et al., Learning Generalizable Skills from Offline Multi-Task Data for Multi-Agent Cooperation. ICLR 2025.
> > ```
> >
> > ---
> > We hope our response addresses your concerns. If so, we wonder if you could kindly consider raising your score? We will also be happy to answer any further questions you may have. Thank you very much!

---

> ### Author Response · Authors · 2025-11-25
> **Looking Forward to Your Feedback**
>
> Dear Reviewer M9Em,
>
> Thank you for your time and feedback! We hope our response and the revision have fully addressed your concerns. If so, we would appreciate it if you could raise your score.
>
> If you have any remaining concerns, we are more than happy to discuss them further.
>
> Best regards,
>
> The Authors

---

### Official Review · Reviewer_NR6Q · 2025-11-08

**Soundness:** 2
**Presentation:** 3
**Contribution:** 2
**Rating:** 4
**Confidence:** 3

**Summary:**

This paper proposes Solo-to-Collaborative RL (SoCo), a framework for leveraging solo (single-agent) demonstrations to accelerate multi-agent reinforcement learning. SoCo pretrains a shared solo policy via behavioral cloning, then adapts it during cooperative training through a policy fusion mechanism combining a gating selector (for choosing among candidate actions) and an action editor (for refining actions via residual corrections). The approach is evaluated on nine cooperative tasks across four scenarios (Spread, LongSwimmer, MultiHalfCheetah, MultiWalker), showing improvements in training efficiency and final performance compared to MATD3 and HATD3 baselines.

**Strengths:**

1. The paper addresses a practically relevant scenario where solo demonstrations are easier to collect than coordinated multi-agent trajectories in domains like collaborative coding, household robotics, and search-and-rescue.

2. The paper is generally well-written and easy to follow.

3. The policy fusion mechanism is intuitive, addressing goal ambiguity through the gating selector and domain shift through the action editor, with a modular design that allows flexibility for different scenarios.

**Weaknesses:**

1. The paper cites PegMARL (Yu et al., 2025), which is incorrectly characterized. According to its abstract, PegMARL utilizes "personalized expert demonstrations" that are "tailored for each individual agent" and "solely pertain to single-agent behaviors without encompassing any cooperative elements," which is functionally identical to what this paper calls "solo demonstrations." This is an important baseline that should be compared experimentally.

2. The approach requires "well-defined, structured, and decomposable" observations (page 4, lines 189-193), which can be restricted: it cannot handle unstructured observations (e.g., raw images, point clouds) and requires manual design of decomposition rules for each environment. Additionally, the method requires tasks to be decomposable into solo subtasks. For example, in a multi-robot box-lifting task, even if observations are decomposable, the task itself is not—a single robot cannot meaningfully lift the box alone. How can solo demonstrations be collected for such fundamentally non-decomposable tasks?

3. The gating selector is defined as g^i_φ : O_i → R^{G_i} (Section 3.3.1, line 220), where G_i is the number of solo views for agent i. However, Gumbel-Softmax requires a fixed output dimension. If different local observations o^i_t correspond to different numbers of solo views, how does the gating selector handle variable output dimensions?

4. Does SoCo only support continuous action spaces? The action editor performs residual adjustments (Equation 4-5), which only makes sense for continuous actions; and all experimental environments seem to use continuous action spaces. If so, this is a significant limitation that should be explicitly stated. If not, how does it work for discrete action spaces?

5. Inference appears computationally expensive, requiring: (1) local observation decomposition, (2) solo policy forward pass, and (3) policy fusion (gating + action editor), rather than a traditional single forward pass through a policy network.

6. The training curves appear to compare only the joint training phase. However, SoCo uses 1M transition samples to pretrain the solo policy, while MATD3 and HATD3 train from scratch. If this pretraining cost is considered, the performance gaps in Figures 2d-2i may not be significant enough.

7. The evaluation environments seem to be relatively simple, mainly particle environments and multi-agent MuJoCo. How would the approach perform on more challenging benchmarks like Google Research Football or SMACv2?

**Questions:**

see Weaknesses.

---

> ### Author Response · Authors · 2025-11-20
> **Response to Reviewer NR6Q (Part 1/3)**
>
> Thank you for your time and effort in reviewing our paper! We are grateful for your constructive suggestions, which have significantly guided our improvements. Please find our responses to your comments below.
>
> ### Weaknesses
> > **Weakness1:** The paper cites PegMARL (Yu et al., 2025), which is incorrectly characterized. According to its abstract, PegMARL utilizes "personalized expert demonstrations" that are "tailored for each individual agent" and "solely pertain to single-agent behaviors without encompassing any cooperative elements," which is functionally identical to what this paper calls "solo demonstrations." This is an important baseline that should be compared experimentally.
>
> Thank you for your detailed comment and suggestion! However, we would like to clarify that the *personalized expert demonstrations* in PegMARL and the *solo demonstrations* in SoCo **are based on fundamentally different assumptions**:
>
> - PegMARL achieves expert guidance mainly through reward reshaping via distribution matching, where the personalized behavior and transition discriminators (Eq. (9)–(10) in [1]) require the personalized observation structure to **be consistent with** that of the cooperative environment.
>
> - In contrast, SoCo explicitly addresses scenarios where solo and cooperative environments **have mismatched observation structures**, making PegMARL’s formulation **inapplicable** to our setting.
>
> Therefore, it is difficult to make a direct comparison between PegMARL and SoCo. To explain this clearly, we have added a more detailed discussion on this distinction in the Appendix F.3.
>
> #### **Reference**
> ```
> [1] Yu, P., Beyond joint demonstrations: Personalized expert guidance for efficient multi-agent reinforcement learning. TMLR 2025.
> ```
>
> > **Weakness2:** The approach requires "well-defined, structured, and decomposable" observations (page 4, lines 189-193), which can be restricted: it cannot handle unstructured observations (e.g., raw images, point clouds) and requires manual design of decomposition rules for each environment. Additionally, the method requires tasks to be decomposable into solo subtasks. For example, in a multi-robot box-lifting task, even if observations are decomposable, the task itself is not—a single robot cannot meaningfully lift the box alone. How can solo demonstrations be collected for such fundamentally non-decomposable tasks?
>
>
> Thank you for your comment! We would like to emphasize that SoCo reveals the **potential of leveraging solo demonstrations from non-cooperative environments** to facilitate cooperative MARL training. We also clarify that the assumption of *well-defined, structured, and decomposable* tasks is common in multi-task (offline) MARL (e.g., [1], [2], and [3]), and it remains **reasonable for a wide range of real-world problems**, such as **network scheduling, code generation, and navigation**, where structured observations and decomposable subtasks naturally exist.
>
> Regarding unstructured observations and the manual design of decomposition rules, one promising direction is to employ **LLMs/VLMs as information processors** [4] to convert raw, unstructured inputs into structured representations suitable for SoCo. However, this extension is **beyond the scope** of the current work.
>
> For inherently non-decomposable tasks, **proxy solo tasks** can be designed to capture relevant individual behaviors. For instance, your *box-lifting* example aligns closely with our **MultiWalker** environment, where two walkers must cooperate to lift a heavy object. As described in **Appendix D**, we collect solo demonstrations by constructing a proxy single-walker task in which the agent lifts a lighter object to learn basic standing and lifting behaviors. As shown in **Figure 2(i)**, even though these solo demonstrations differ significantly in dynamics from the cooperative task, **SoCo still substantially improves the backbone algorithm’s training efficiency**.
>
> We have added these discussion in Appendix F.2.
>
> #### **References**
> ```
> [1] Hu, S. et al., UPDeT: Universal Multi-agent RL via Policy Decoupling with Transformers. ICLR 2021.
> [2] Zhang, F. et al., Discovering Generalizable Multi-agent Coordination Skills from Multi-task Offline Data. ICLR 2023.
> [3] Liu, S. et al., Learning Generalizable Skills from Offline Multi-Task Data for Multi-Agent Cooperation. ICLR 2025.
> [4] Cao, Y., Survey on Large Language Model-Enhanced Reinforcement Learning: Concept, Taxonomy, and Methods. TNNLS 2025.
> ```

---

> ### Author Response · Authors · 2025-11-20
> **Response to Reviewer NR6Q (Part 2/3)**
>
> > **Weakness3:** The gating selector is defined as g^i_φ : O_i → R^{G_i} (Section 3.3.1, line 220), where G_i is the number of solo views for agent i. However, Gumbel-Softmax requires a fixed output dimension. If different local observations o^i_t correspond to different numbers of solo views, how does the gating selector handle variable output dimensions?
>
> Thank you for your comment! When $G_i$ varies across different observations, a feasible solution is to **reuse the same gating network by modifying its input structure** rather than changing the output dimension. Specifically, instead of producing a vector over all solo views at once, the gating selector can take both the **local observation** $o^i_t$ and each **solo view $\tilde{o}^{i,k}_t$ (or action candidate $a^{i,k}_t$)** as joint inputs, and output a scalar weight $w^{i,k}_t = g^i_φ(o^i_t, \tilde{o}^{i,k}_t)$ for each candidate. These weights can then be collected and normalized via Gumbel–Softmax sampling, which still allows end-to-end gradient propagation. This formulation can effectively handle variable $G_i$ without requiring a fixed output dimension. We have added a remark about this in Section 3.3.1.
>
> > **Weakness4:** Does SoCo only support continuous action spaces? The action editor performs residual adjustments (Equation 4-5), which only makes sense for continuous actions; and all experimental environments seem to use continuous action spaces. If so, this is a significant limitation that should be explicitly stated. If not, how does it work for discrete action spaces?
>
> Thank you for your suggestion! We acknowledge that while the idea behind SoCo is indeed insightful, our current work only explicitly demonstrates its effectiveness in continuous action spaces. However, **the intrinsic characteristics of discrete-action spaces make a direct extension of SoCo non-trivial**. We have **explicitly added** this limitation and future direction in Conclusion, and added a discussion in Appendix F.1.
>
> Specifically, we would like to share several insights into the challenges involved:
>
> - A straightforward implementation would apply SoCo at the logits level. However, since discrete actions rely on **argmax-based sampling**, fine-tuning logits to adjust the final action is extremely difficult. For instance, in our preliminary attempts on the *SMAC-v2 Protoss 8v8* environment, early-stage coordination policy tried to adjust over 50% of actions, yet fewer than 10% of those adjustments successfully changed the executed actions. The mismatch between intended and executed actions limits the exploration, and the alignment only appeared at later training stages.
>
> - Moreover, the continuous-control MARL tasks we study, though appearing simpler, **actually provide more optimization headroom and clearer insight into SoCo’s effect on coordination efficiency.** In contrast, we have observed that existing MARL algorithms, such as QMIX-style methods and MAPPO, already achieve very high efficiency and performance in most discrete-action benchmarks (e.g., SMAC-v1/-v2, Google Research Football) under implementations like MAPPO's official implementation[1], PyMARL2[2], PyMARL3[3] and HARL[4]. Although these tasks seem to be "more difficult", their efficiency gap compared with SoCo’s “intention-matching process” could be minimal, leaving little room for SoCo to bring further improvement. **Establishing more complex discrete-action tasks remains an important direction for MARL community.**
>
> #### **References**
> ```
> [1] Yu, C., The Surprising Effectiveness of PPO in Cooperative Multi-Agent Games. NeurIPS 2022.
> [2] Hu, J. et al., Rethinking the Implementation Tricks and Monotonicity Constraint in Cooperative Multi-Agent Reinforcement Learning. ArXiv 2021, ICLR 2023 Blog Track.
> [3] Hao, X. et al., Boosting Multi-Agent Reinforcement Learning via Permutation Invariant and Permutation Equivariant Networks. ICLR 2023.
> [4] Zhong, Y. et al., Heterogeneous-agent reinforcement learning. JMLR 2024.
> ```

---

> > ### Author Response · Authors · 2025-11-20
> > **Response to Reviewer NR6Q (Part 3/3)**
> >
> > > **Weakness5:** Inference appears computationally expensive, requiring: (1) local observation decomposition, (2) solo policy forward pass, and (3) policy fusion (gating + action editor), rather than a traditional single forward pass through a policy network.
> >
> > Thank you for your comment! We would like to clarify that the **local observation decomposition** in SoCo is **rule-based**, and the **coordination policy** in the action editor shares the same architecture as the backbone MARL algorithm. Therefore, the only additional *computationally intensive* components are **the solo policy forward pass** and **the gating module**.
> >
> > It is also important to note that, although SoCo introduces extra computation to leverage solo demonstrations, our experiment results in Figure 2 show that in **complex tasks where traditional single-pass MARL models train inefficiently**, SoCo **significantly accelerates convergence in terms of interaction steps**. Hence, this trade-off is justified by the overall efficiency gain. Nevertheless, we agree that exploring **lighter-weight extensions** of SoCo is an interesting direction for future work.
> >
> > > **Weakness6:** The training curves appear to compare only the joint training phase. However, SoCo uses 1M transition samples to pretrain the solo policy, while MATD3 and HATD3 train from scratch. If this pretraining cost is considered, the performance gaps in Figures 2d-2i may not be significant enough.
> >
> > Thank you for your detailed comment! We would like to emphasize that the **pretraining cost does not affect the evaluation of SoCo’s training performance** for the following reasons:
> >
> > - The solo policy pretraining is conducted **entirely offline**, without any online environment interaction. Similar to *offline-to-online RL* settings (e.g., [1], [2], and [3]), such offline pretraining costs are **typically excluded** when evaluating online training efficiency and performance.
> >
> > - The pretrained **solo policies are reusable** across similar cooperative tasks. For example, once the 2×2 Swimmer solo policy is pretrained, we can directly load it for the cooperative training in **Figures 2(d–f)** without retraining. In contrast, **MATD3** and **HATD3** must always be trained from scratch.
> >
> > - In terms of **wall-clock time**, the pretraining process takes only about **30–60 minutes**, which is negligible compared to the cooperative training phase.
> >
> > #### **References**
> > ```
> > [1] Seunghyun Lee, Younggyo Seo, Kimin Lee, Pieter Abbeel, Jinwoo Shin. Offline-to-Online Reinforcement Learning via Balanced Replay and Pessimistic Q-Ensemble. CoRL 2022.
> > [2] Mitsuhiko Nakamoto, Yuexiang Zhai, Anikait Singh, Max Sobol Mark, Yi Ma, Chelsea Finn, Aviral Kumar, Sergey Levine. Cal-QL: Calibrated Offline RL Pre-Training for Efficient Online Fine-Tuning. NeurIPS 2023.
> > [3] Hai Zhong, Xun Wang, Zhuoran Li, Longbo Huang. Offline-to-Online Multi-Agent Reinforcement Learning with Offline Value Function Memory and Sequential Exploration. AAMAS 2025.
> > ```
> >
> > > **Weakness7:** The evaluation environments seem to be relatively simple, mainly particle environments and multi-agent MuJoCo. How would the approach perform on more challenging benchmarks like Google Research Football or SMACv2?
> >
> > Thank you for your comment! As mentioned in our response to **Weakness 4**, we would like to clarify that the **continuous-control tasks** we consider are **far from trivial**. As shown in the training curves, environments such as **Spread-5** and **MultiWalker** remain highly challenging even for state-of-the-art continuous-action MARL algorithms.
> >
> > On the other hand, extending SoCo to **discrete-action environments** such as Google Research Football or SMACv2 is **non-trivial**, as discussed earlier. Moreover, although these benchmarks appear complex, existing  implementation of MARL algorithms for discrete action spaces (e.g., QMIX-style, MAPPO) already achieve **near-saturated performance**, leaving limited room for measurable improvement. Designing **harder discrete-action tasks** that can meaningfully evaluate new methods remains an open problem and a valuable direction for future work.
> >
> > ---
> > We hope our response addresses your concerns. If so, we wonder if you could kindly consider raising your score? We will also be happy to answer any further questions you may have. Thank you very much!

---

> ### Author Response · Authors · 2025-11-25
> **Looking Forward to Your Feedback**
>
> Dear Reviewer NR6Q,
>
> Thank you for your time and feedback! We hope our response and the revision have fully addressed your concerns. If so, we would appreciate it if you could raise your score.
>
> If you have any remaining concerns, we are more than happy to discuss them further.
>
> Best regards,
>
> The Authors

---

> > ### Comment · Reviewer_NR6Q · 2025-11-26
> >
> > Here are a couple of follow-up clarifications from my side.
> >
> > First, regarding the distinction from PegMARL: I believe there is a misunderstanding in the rebuttal. PegMARL does not require the personalized task to share the same full observation structure as the cooperative environment. They only assume that “the state space Si and action space Ai of the personalized task Ti are the same as the local state and action spaces of agent i from the joint task”. In their partially observable experiments, PegMARL handles observation mismatch simply by removing dimensions that are not observable in the personalized MDP before feeding them to the discriminators. This is essentially the same kind of preprocessing that SoCo performs -- your method also requires observations to be decomposable, and you reconstruct solo views via concatenation, zero-padding, and similar operations. So the claim that PegMARL is inapplicable due to observation-structure inconsistency is not accurate; both methods rely on very similar assumptions about how joint observations can be mapped into agent-specific solo views.
> >
> > Second, even after the rebuttal, the overall method still feels quite restricted. It depends heavily on manually decomposable observations and tasks where meaningful solo subtasks (or proxy solo tasks) can be constructed; and it currently only works for continuous action spaces.

---

> > > ### Author Response · Authors · 2025-12-02
> > > **Response to Official Comment by Reviewer NR6Q (Part 1/2)**
> > >
> > > Thank you for the clarification! Please find our further response below:
> > > > First, regarding the distinction from PegMARL: I believe there is a misunderstanding in the rebuttal. PegMARL does not require the personalized task to share the same full observation structure as the cooperative environment. They only assume that “the state space Si and action space Ai of the personalized task Ti are the same as the local state and action spaces of agent i from the joint task”. In their partially observable experiments, PegMARL handles observation mismatch simply by removing dimensions that are not observable in the personalized MDP before feeding them to the discriminators. This is essentially the same kind of preprocessing that SoCo performs -- your method also requires observations to be decomposable, and you reconstruct solo views via concatenation, zero-padding, and similar operations. So the claim that PegMARL is inapplicable due to observation-structure inconsistency is not accurate; both methods rely on very similar assumptions about how joint observations can be mapped into agent-specific solo views.
> > >
> > > We apologize for our earlier misunderstanding of PegMARL and have corrected the relevant descriptions in the revised manuscript. However, after carefully revisiting the PegMARL paper [1] and its released code, **we still find that PegMARL does not fully fit our setting**:
> > >
> > > - First, although PegMARL also uses a decomposer to extract a solo observation, it still **cannot handle cases where a single cooperative observation corresponds to multiple solo views** (e.g., our Spread scenario).
> > > - Second, PegMARL mainly achieves guidance via individual reward shaping, which makes it more naturally suited to fully decentralized methods such as independent PPO, and **less straightforward to apply to standard CTDE methods** such as MATD3 [2] and HASAC [3] that rely on a centralized Q-function and a shared team reward. The PegMARL paper itself notes that it uses *“separate policy and critic networks, discriminators, and optimizers”* (Appendix C.1, p.17 in [1]), and the official implementation also suggests that PegMARL is closer to a *fully decentralized* approach. By contrast, SoCo is **primarily built upon the CTDE paradigm**, so the two methods are **not directly comparable** under our setting.
> > > - Finally, although PegMARL claims to be applicable to “*discrete and continuous environments*,” in practice all the evaluated tasks involved in [1] have **discrete action spaces**. This is explicitly stated in the Appendix A (p.14) and Table 1 in Appendix B.1 (p.15) of [1], and the released code indeed only supports discrete action spaces. In contrast, **SoCo focuses mainly on continuous-action settings**, which, as discussed in our Appendix F, are substantially different from discrete-action ones and **still leave significant room for improvement**; we therefore believe our study remains **highly  valuable**.
> > >
> > > Nevertheless, we still port PegMARL to our tasks for comparison with SoCo, and we have added the corresponding results in Figure 2. Concretely, we retain PegMARL’s decentralized nature but modify the inputs and outputs of the actor–critic and discriminator to support continuous action spaces, and in scenarios with multiple solo views we randomly select one as the personalized observation. **The results show that SoCo significantly outperforms PegMARL.** That said, we view SoCo and PegMARL as orthogonal rather than competing techniques, and we believe they could be fruitfully combined in future work.
> > >
> > > #### **References**
> > > ```
> > > [1] Yu, P. et al., Beyond Joint Demonstrations: Personalized Expert Guidance for Efficient Multi-Agent Reinforcement Learning. TMLR 2025.
> > > [2] Ackermann, J. et al., Reducing overestimation bias in multi-agent domains using double centralized critics, 2019.
> > > [3] Liu, J. et al., Maximum Entropy Heterogeneous-Agent Reinforcement Learning. ICLR 2024.
> > > ```

---

> > > > ### Author Response · Authors · 2025-12-02
> > > > **Response to Official Comment by Reviewer NR6Q (Part 2/2)**
> > > >
> > > > > Second, even after the rebuttal, the overall method still feels quite restricted. It depends heavily on manually decomposable observations and tasks where meaningful solo subtasks (or proxy solo tasks) can be constructed; and it currently only works for continuous action spaces.
> > > >
> > > > Regarding the concern that our method still feels “quite restricted,” we would like to clarify that **SoCo is intended as an insightful and general framework for leveraging solo demonstrations to accelerate cooperative MARL training via a gating selector and an action editor**, rather than a narrowly tailored heuristic. The assumption of a decomposed observation is in fact **standard** in many multi-task and cooperative MARL settings (e.g., [1][2][3][4]), and, as we discussed earlier for the box-carrying example, non-directly decomposable tasks can often be handled by constructing proxy solo tasks. While we would certainly welcome concrete counterexamples where “meaningful solo subtasks (or proxy solo tasks)” cannot be defined, our current evaluation **already covers a broad class of tasks**. For more general settings, Appendix F discusses how one might combine SoCo with powerful representation learners (e.g., LLMs) to automatically induce such structures, but this direction **goes beyond the scope** of the present work and should not diminish the contribution made by SoCo itself.
> > > >
> > > > As for the comment that SoCo “currently only works for continuous action spaces,” we would like to clarify the following.
> > > >
> > > > - First, as discussed in Appendix F, the continuous-action settings we focus on **differ substantially from discrete-action ones**, and a direct extension is **nontrivial**. We view the extension for discrete actions as an important avenue for future work.
> > > >
> > > > - Moreover, continuous-action cooperative control **still has ample room for improvement**, which further **highlights the relevance and value of SoCo**.
> > > >
> > > > - Finally, we emphasize that **effectively exploiting solo demonstrations is a new and largely open problem**, and **it is unrealistic for a single work to comprehensively solve all variants** (e.g., CTDE vs fully decentralized, continuous vs discrete actions). For example, PegMARL [1] is an important step in this direction but is designed for *fully decentralized, discrete-action settings*, whereas SoCo targets *CTDE algorithms in continuous action spaces*. Our experiments show that, in the latter regime, **SoCo substantially outperforms PegMARL**.
> > > >
> > > > Thus, while we agree that extending SoCo to additional settings (such as discrete actions) is valuable future work, this limitation does not undermine the significance of the framework and the contributions made in the current paper.
> > > >
> > > > #### **References**
> > > > ```
> > > > [1] Yu, P. et al., Beyond Joint Demonstrations: Personalized Expert Guidance for Efficient Multi-Agent Reinforcement Learning. TMLR 2025.
> > > > [2] Hu, S. et al., UPDeT: Universal Multi-agent RL via Policy Decoupling with Transformers. ICLR 2021.
> > > > [3] Zhang, F. et al., Discovering Generalizable Multi-agent Coordination Skills from Multi-task Offline Data. ICLR 2023.
> > > > [4] Liu, S. et al., Learning Generalizable Skills from Offline Multi-Task Data for Multi-Agent Cooperation. ICLR 2025.
> > > > ```

---

### Official Review · Reviewer_qeTh · 2025-11-10

**Soundness:** 2
**Presentation:** 3
**Contribution:** 3
**Rating:** 4
**Confidence:** 3

**Summary:**

The authors propose to collect trajectories from the single-agent counterparts of multi-agent tasks, use behavior cloning (offline reinforcement learning) to first obtain a shared prior policy $\pi_{\text{solo}}$, and then let each multi-agent actor fine-tune based on $\pi_{\text{solo}}$ to train on the multi-agent tasks. The authors demonstrate the superiority of their method on many continuous tasks.

**Strengths:**

The paper takes a novel perspective, proposing to use a more general single-agent task corresponding to a multi-agent task as a kind of initialization for the multi-agent problem.

It also shows good empirical performance, requiring fewer training steps within the multi-agent environments.

**Weaknesses:**

**Shortcomings in the comparative experiments:**

1. Since the method is positioned as a “plug-in,” it should be combined with more backbones to demonstrate extensibility. For the authors’ continuous-action settings, baselines such as MADDPG and SAC should be included.

2. The paper claims the method can extend to discrete-action environments, yet all reported experiments are on continuous control. The authors should add concrete results on discrete-action tasks.

3. I would like to see ablations on the action candidates—for example, a setting that uses only the agent’s own decomposed observation ooo as the **sole** action candidate.

4. In a sense, the method reuses a single-agent initialization for multi-agent models. It is understandable that this leads to significantly faster convergence than the baselines. However, the **final** performance improvements appear less pronounced (although accelerating convergence is itself a strong advantage). I would like to see, for Figure 2 (g, h, i), how much improvement remains when training to full convergence.

**Questions:**

see weakness.

---

> ### Author Response · Authors · 2025-11-20
> **Response to Reviewer qeTh**
>
> Thank you for your time and effort in reviewing our paper! We are grateful for your constructive suggestions, which have significantly guided our improvements. Please find our responses to your comments below.
>
> ### Weaknesses
>
> > **Weakness1:** Since the method is positioned as a “plug-in,” it should be combined with more backbones to demonstrate extensibility. For the authors’ continuous-action settings, baselines such as MADDPG and SAC should be included.
>
> Thank you for your suggestion! We have added additional experiments in Appendix E based on HASAC[1], **a recently proposed and advanced stochastic-policy** MARL algorithm.
>
> The results show that, for a recent advanced stochastic-policy algorithm, SoCo can still **consistently accelerate multi-agent training and achieve competitive or even superior final performance** compared to the baseline. This further demonstrates the plug-and-play nature of SoCo and its strong potential to leverage solo demonstrations to improve cooperative training.
>
> #### **Reference**
> ```
> [1] Liu, J. et al., Maximum Entropy Heterogeneous-Agent Reinforcement Learning. ICLR 2024.
> ```
>
> > **Weakness2:** The paper claims the method can extend to discrete-action environments, yet all reported experiments are on continuous control. The authors should add concrete results on discrete-action tasks.
>
> Thank you for your suggestion! We would like to clarify that we did **not claim** our method can be directly extended to discrete-action environments. In fact, as we mentioned in our revised discussion section, while the idea behind SoCo is indeed insightful, **the intrinsic characteristics of discrete-action spaces make such an extension non-trivial**.
>
> Specifically, we would like to share several insights into the challenges involved:
>
> A straightforward implementation would apply SoCo at the logits level. However, since discrete actions rely on **argmax-based sampling**, fine-tuning logits to adjust the final action is extremely difficult. For instance, in our preliminary attempts on the *SMAC-v2 Protoss 8v8* environment, early-stage coordination policy tried to adjust over 50% of actions, yet fewer than 10% of those adjustments successfully changed the executed actions. The mismatch between intended and executed actions limits the exploration, and the alignment only appeared at later training stages.
>
> Following your suggestion, we have explicitly added a discussion of this limitation and future directions in the revision. See Page 10, line 526, and Appendix F.1.
>
> > **Weakness3:** I would like to see ablations on the action candidates—for example, a setting that uses only the agent’s own decomposed observation ooo as the sole action candidate.
>
> Thank you for your comment. Could you please clarify what you mean more specifically?
>
> > **Weakness4:** In a sense, the method reuses a single-agent initialization for multi-agent models. It is understandable that this leads to significantly faster convergence than the baselines. However, the final performance improvements appear less pronounced (although accelerating convergence is itself a strong advantage). I would like to see, for Figure 2 (g, h, i), how much improvement remains when training to full convergence.
>
>
> Thank you for your comment and for acknowledging SoCo’s advantage in accelerating training! We would like to emphasize that SoCo primarily enhances the training efficiency of backbone algorithms by **reducing ineffective exploration through solo demonstrations**. Consequently, on tasks where the backbone already explores effectively (e.g., **MAMuJoCo**), the final performance is expected to remain comparable to training from scratch.
>
> For the fully converged performance, **we have extended Figure 2(i) to 10M steps.** As for Figure 2(g, h) (the two MAMuJoCo tasks), results are uniformly reported up to 2 M steps since the MuJoCo simulator frequently crashes due to numerical instability from dynamic tendon coupling, **a simulation issue affecting both the backbone baseline and the SoCo variant**. Nevertheless, current trends indicate that their final performance would remain **comparable even better upon full convergence**.
>
> ---
> We hope our response addresses your concerns. If so, we wonder if you could kindly consider raising your score? We will also be happy to answer any further questions you may have. Thank you very much!

---

> ### Author Response · Authors · 2025-11-25
> **Looking Forward to Your Feedback**
>
> Dear Reviewer qeTh,
>
> Thank you for your time and feedback! We hope our response and the revision have fully addressed your concerns. If so, we would appreciate it if you could raise your score.
>
> If you have any remaining concerns, we are more than happy to discuss them further.
>
> Best regards,
>
> The Authors

---

### Author Response · Authors · 2025-12-02
**General Response (Part 1/2)**

We sincerely thank the PC, SAC, AC, and all reviewers for their time and thoughtful feedback! Their comments have helped us substantially improve the clarity and quality of the paper.

---
### 1. Summary of our contributions
- **Underexplored problem**\
We study the underexplored problem of leveraging solo demonstrations for cooperative MARL, and show that such data, despite lacking explicit cooperative signals, can substantially accelerate multi-agent training.
- **Novel framework**\
We propose SoCo, a novel solo-to-cooperative transfer framework that reuses a shared solo policy via decomposed local observations and a policy fusion module combining a learnable gating selector and action editor.
- **Empirical validation**\
We validate SoCo on diverse cooperative benchmarks, demonstrating improved sample efficiency and strong performance under observation ambiguity and domain shift, highlighting solo demonstrations as a scalable resource for MARL.
### 2. Main changes in the revised manuscript
Following the reviewers’ suggestions, we have made the following key revisions:
- **Clarifications of assumptions and scope**
 We clarify that SoCo is primarily designed for **off-policy CTDE MARL with continuous actions** (Lines 120, 144), and we explicitly discuss its limitations and extensions:
     - Add future directions to the conclusion (p.10, Lines 537–539)
     - Provide detailed discussion of SoCo’s applicability in Appendix F (pp. 23–24), covering (i) continuous vs. discrete actions (F.1), (ii) unstructured or non-decomposable problems (F.2), (iii) differences from PegMARL and the difficulty of direct comparison (F.3), and (iv) applicability to on-policy backbones (F.4).
- **Additional baselines and experiments**
    - Extended the MultiWalker task to 10M environment steps (p.7, Line 338).
    - Add MAPPO and PegMARL as additional baselines to enable a cross-method evaluation of SoCo (p.7, Lines 327–333; Figure 2, p.8). We also clarify the purpose and fairness of these comparisons (p.8, Lines 412–421). The new results show that **SoCo still maintains a clear advantage over these methods**.
    - Add an implementation and experiments with HASAC as the backbone, showing that SoCo can also be **successfully applied to stochastic-policy algorithms** (Appendix E, pp. 21–23).
    - Include experiments using MAPPO as the backbone, illustrating both the applicability of SoCo to on-policy methods and the associated challenges (Figure 8 in Appendix F.4, p. 24).
    - Add experiments on the impact of different solo demo qualities on SoCo, and **uncover a new insight**: Solo demo quality does not monotonically predict cooperative benefit. (Sec. 4.4, p. 10).
- **Improved explanations**
  - A concrete example to illustrate the observation decomposer (Appendix D.2.3, p.20).
  - An explanation of how to compute gating weights when candidates differ (Remark on p.5, Lines 243–245).
  - Implementation details on how PegMARL is adapted to our experimental environments (Appendix D.3, p.21, Lines 1083–1091).
- **Writing and notation fixes**
  - Correct our description of the related work PegMARL and clarified its relationship and differences with SoCo (Lines 75–79, p.2; Lines 521–524, p.10; Appendix B, Lines 873–878, p.17; Appendix F.3, p.24).
  -  Unified the starting index of the solo view between Figure 1 and the main text (Line 203, p.4).

We explicitly mark all substantial changes in red in the revised version.

---

> ### Author Response · Authors · 2025-12-02
> **General Response (Part 2/2)**
>
> ### 3. Responses to main concerns
> Below, we summarize how we address the main concerns.
> - **Limited Baselines and Experiments** (qeTh, NR6Q, M9Em, fw46, and wccp)
>     - Regarding the experimental environments, we have clarified in the rebuttal that the nine *continuous-action* tasks we consider are **in fact quite challenging for existing algorithms**, whereas seemingly more complex discrete-action benchmarks (e.g., SMAC) have been extensively studied and are close to saturation.
>     - As for the relatively limited set of baselines in the original submission, the revised version includes **a richer set of methods**: we add PegMARL and MAPPO for *cross-method evaluation*, incorporate results with recent advanced HASAC as a *stochastic-policy backbone* for SoCo. The experiments show that **SoCo still achieves better performance**. We also further explore experiments using MAPPO as the *on-policy backbone* for SoCo.
> - **Comparison with PegMARL** (NR6Q and M9Em)\
> In both the rebuttal and the revised manuscript, we provide an in-depth discussion of PegMARL, explaining that it is more suitable for *fully decentralized algorithms in discrete-action settings* and *cannot handle scenarios with multiple solo views*. In contrast, SoCo is designed for *continuous-action CTDE algorithms*, making the two methods **orthogonal** and **not directly comparable**. Our cross-method experiments further show that, under our setting, **SoCo significantly outperforms PegMARL**.
> - **Extension to Discrete Action Spaces** (qeTh, NR6Q, and fw46)\
> First, we clarify that SoCo **primarily targets continuous-action settings**, and extension to discrete action is nontrivial. As discussed in the rebuttal and Appendix F, a naive logits-level adaptation to suffers from a severe action mismatch, and many standard discrete-action MARL benchmarks (e.g., SMAC) are already **close to saturation** under strong baselines (QMIX-style methods, MAPPO, etc.), leaving limited headroom for SoCo-style improvements. Thus, we consider it more appropriate to first develop more challenging, unsaturated discrete-action benchmarks before pursuing a principled discrete extension of SoCo.
> - **Extension to Unstructured or Non-Decomposable Tasks** (NR6Q, M9Em and wccp)\
> In the rebuttal, we emphasize that assuming structural and decomposable tasks is a **common and reasonable** setting in prior work [1–3], and already covers a **substantial** subset of practical scenarios. Truly unstructured or non-decomposable tasks are **beyond the scope** of this paper. Nevertheless, we proactively discuss possible extensions in the rebuttal and Appendix F, such as leveraging vLMs/LLMs to parse unstructured inputs and constructing proxy solo tasks to handle non-decomposable settings.
>
> **References**
> ```
> [1] Hu, S. et al., UPDeT: Universal Multi-agent RL via Policy Decoupling with Transformers. ICLR 2021.
> [2] Zhang, F. et al., Discovering Generalizable Multi-agent Coordination Skills from Multi-task Offline Data. ICLR 2023.
> [3] Liu, S. et al., Learning Generalizable Skills from Offline Multi-Task Data for Multi-Agent Cooperation. ICLR 2025.
> ```
> ---
> We thank the PC, SAC, and AC for considering our work. We hope our rebuttal addresses the main concerns raised in the reviews, and the paper can be recommended for acceptance.

---

### Meta-Review · Area_Chair_V7GV · 2026-01-07

**Summary:**

This paper was reviewed by five reviewers and received five negative recommendations. The major concerns were
- insufficient experiments
- lack of comparisons with other baseline methods
- relying on restricted observations.

**Reviewer Concerns:**

Although the authors  have provided detailed responses to the comments, the concerns about lack of comparisons with other baseline methods and relying on  restricted observations are still outstanding.  After carefully reading the comments and responses, the AC think this paper cannot be accepted in its current form.

**Reviewer Scores:**

Given that all reviewers initially recommended acceptance and that there likely would be outstanding concerns, it is highly unlikely that the reviewers would have sufficiently changed their scores to warrant acceptance.

---

### Decision · Program_Chairs · 2026-01-26

Reject